# PolicyGuard: Towards Test-time and Step-level Adversary Defense for Reinforcement Learning Agent

**Junfeng Guo** [1]   **Heng Huang** [1]

## Abstract

While real-world applications of reinforcement learning (RL) are becoming increasingly popular, the security of RL systems deserve more attention and exploration. In particular, recent work has revealed that RL agents are vulnerable to backdoor attacks, where a victim agent behaves normally under standard conditions but executes malicious actions when a specific trigger is activated. Existing backdoor defenses for RL either require access to the agent's internal parameters, operate only at the model or trajectory level, or are limited to specific attack types. To ensure the security of RL agents, we propose `PolicyGuard`, a *test-time step-level* backdoor defense which leverages Gaussian Process (GP) posterior variance and adapts pseudo trajectories to enable uncertainty computation for individual time step. Besides, we also provide theoretical foundations to explain the efficacy of GP posterior variance. Extensive experiments across seven RL games demonstrate that PolicyGuard achieves state-of-the-art detection performance in most cases, with average AUROC of 0.856 for perturbation-based attacks and 0.859 for adversary-agent attacks.

## 1. Introduction

Reinforcement Learning (RL) is proposed to train intelligent agents that take actions to maximize cumulative rewards in a given environment (Silver et al., 2021). Such incentive-driven properties have enabled RL to achieve remarkable success across diverse applications, including game playing (Mnih, 2013), robotics (Singh et al., 2022), autonomous driving (Sallab et al., 2017), and traffic control (Rasheed et al., 2020). Given that many real-world RL applications are safety-critical (Cheng et al., 2019; Singh et al., 2022; Sal-

lab et al., 2017), it becomes increasingly important to ensure the security and robustness of RL agents before deployment.

Recent studies have revealed that RL systems are vulnerable to various types of backdoor attacks (Wang et al., 2021; Kiourti et al., 2020) . Depending on the trigger mechanism and attack scenario, existing backdoor attacks can be categorized into two types: *perturbation-based* (Kiourti et al., 2020) and *adversary-agent* (Wang et al., 2021) attacks. Perturbation-based attacks directly embed a specific pixel pattern as a trigger into the victim agent's observation state, and are widely applied in single-agent environments such as Atari games (Mnih, 2013). In contrast, adversary-agent attacks do not explicitly inject triggers into the observation space; instead, the trigger is formed through the complex dynamics between an adversary agent's specific actions and the victim agent's state, making them particularly effective in multi-players environments. Both types of attacks can cause catastrophic failures when the trigger is activated, severely compromising the reliability of RL systems.

To ensure the security of RL agents against malicious backdoors in the practical settings (black-box), in this work, we consider the problem of *test-time backdoor defense* for RL, which aims at identifying potentially backdoor-infected time steps during deployment and preventing the execution of compromised behaviors. This problem poses great challenges compared to backdoor defense in supervised learning (Dong et al., 2021; Wang et al., 2019) or at the model/trajectory level (Chen et al., 2023; Yuan et al., 2024; Bharti et al., 2022), for several reasons. *i)* unlike the stateless trigger detection in image classifiers (Gao et al., 2019; Guo et al., 2023b; Wang et al., 2019), RL backdoor defense must reason about temporal dependencies across sequential state-action pairs (Yuan et al., 2024; Guo et al., 2023a). *ii)* in practical deployment scenarios, the defender typically only has black-box access to the agent without knowledge of its internal parameters (Bharti et al., 2022; Chen et al., 2023) or value network (Chen et al., 2023). *iii)* existing defenses require future time steps before making detection decisions (Yuan et al., 2024; Anonymous, 2025), or are designed specifically for one type of attack (Guo et al., 2023a; Bharti et al., 2022) , which limits their applicability and generalization for test-time step-level backdoor defense.

---

[1]Computer Science, University of Maryland College Park, USA.

*Proceedings of the 43rd International Conference on Machine Learning*, Seoul, South Korea. PMLR 306, 2026. Copyright 2026 by the author(s).

To address these challenges, we propose `PolicyGuard`, a test-time and step-level backdoor defense approach that can detect potentially compromised time steps *online* without accessing the agent's internal parameters and entire episodes. The key insight of PolicyGuard is that Gaussian Process (GP) posterior variance (Wenger et al., 2022) built with the suspicious model's normal trajectories naturally quantifies epistemic uncertainty; when an agent exhibits backdoor-triggered behaviors, these trajectories deviate from state-actions observed during normal operation, resulting in elevated posterior variance. Specifically, `PolicyGuard` first collects state-action trajectories from a clean environment and trains an additive GP model (Guo et al., 2021) to capture normal behavioral patterns. During deployment, for each incoming state-action pair, `PolicyGuard` constructs pseudo trajectories to compute context-aware posterior variances, which serve as uncertainty scores to distinguish backdoor-infected steps from benign ones.

Evidenced by extensive experiments across seven RL games (Mnih, 2013; Bansal et al., 2017) and different types of attacks, `PolicyGuard` achieves superior detection performance compared to existing defense methods in most cases. Notably, under our considered hard-coded attack settings where optimization-based defenses (*i.e.,* Neural Cleanse (Wang et al., 2019), PolicyCleanse (Guo et al., 2023a)) completely fail (yielding AUROC scores near 0.5), `PolicyGuard` maintains strong detection performance (0.868 for perturbation-based and 0.878 for adversary-agent attacks), demonstrating its efficacy in practical black-box scenarios. We also evaluate `PolicyGuard` under various ablation settings, including different trigger sizes, trigger action lengths, and adaptive attacks, *etc*. We summarize our contributions as follows:

- We propose `PolicyGuard`, a test-time and step-level backdoor defense approach for RL that leverages GP posterior variance for uncertainty quantification.

- We provide theoretical analyses that establish the discriminability of the posterior variance of GP between the backdoor-infected and benign state-action pairs and show its effectiveness in our experimental studies. We further propose pseudo trajectories to enable uncertainty computation for individual time steps.

- We conduct comprehensive experiments across diverse environments and attack strategies, demonstrating `PolicyGuard` achieves state-of-the-art detection performance in most cases and practical scenarios.

### 1.1. Background and Related Work

**Backdoor Attacks Against RL.** Recent studies have shown that RL is prone to various backdoor attacks (Kiourti et al., 2020; Wang et al., 2021). According to previous work, existing backdoor attacks can be categorized into *perturbation-based attacks* (Kiourti et al., 2020) and *adversary-agent attacks* (Wang et al., 2021) based on trigger patterns and attack strategies. Perturbation-based attacks use a specific pixel pattern as the trigger and directly embed the trigger into the victim agent's observation state. As for adversary-agent attacks, the trigger pattern does not explicitly appear in the victim agent's observation state but appears as the dynamics between an adversary agent's specific (trigger) action and the victim agent's state. Perturbation-based attacks are widely applied in the single-agent environment; while adversary-agent attacks perform effective in the multi-player environment (Bansal et al., 2017).

**Backdoor Defense for RL.** We focus on the testing-phase backdoor defense, which is widely studied in the supervised classification tasks (Guo et al., 2023b; Gao et al., 2019; Li et al., 2021). Unfortunately, due to the dramatic and fundamental difference between supervised classification task and RL, these techniques have limited efficacy in RL. Beyond that, most existing work either have a different threat model (Yuan et al., 2024; Chen et al., 2023; Bharti et al., 2022) from ours or are limited in practicability (Anonymous, 2025) and generalizability (Guo et al., 2023a). Specifically, the defenses proposed in (Chen et al., 2023) require access the agent's value network . (Bharti et al., 2022) and (Guo et al., 2023a) is designed only for perturbation-based or adversary-agent attacks. To our best knowledge, SHINE (Yuan et al., 2024) is the only existing work shown effective for testing-phase backdoor defense for RL in a generalized setting. However, SHINE (Yuan et al., 2024) still require complete an entire episode before identifying potentially backdoor-infected steps, which demonstrates limitation in online backdoor defenses.

## 2. Preliminaries

In this section, we will describe the problem formulations, threat model considered, and key challenges to identify backdoor-infected behaviors under our scenarios.

### 2.1. Reinforcement Learning

Reinforcement learning (RL) can be technically formulated as a Markov Decision Process (MDP), which consists of a sequence of state ($\mathcal{S}$), actions ($\mathcal{A}$), transition function ($\mathcal{T}$), and reward ($r$), *i.e.*, $\mathcal{M} := (\mathcal{S}, \mathcal{A}, \mathcal{T}, R)$. $\mathcal{T}$ is the transition function conditioned on states ($\mathcal{S}$) and actions ($\mathcal{A}$), *i.e.*, $\mathcal{S} \times \mathcal{A} \rightarrow \mathcal{S}$. We define the reward function $R$ as $\mathcal{S} \times \mathcal{A} \times \mathcal{S} \rightarrow \mathbb{R}$. For simplicity, in multi-agent environments (Bansal et al., 2017), we fix the policy of the opponent agents other than the target agent, and the environment for the target agent becomes an MDP. The goal of a RL policy $\pi(\cdot|\theta)$ parameterized by $\theta$ is to maximize the accumulated rewards as follows:

$$\theta^* = \arg\max_\theta \sum_{t=0}^{\infty} \gamma_t R(s_t, a_t, s_{t+1}), \qquad (1)$$

where $a_t \sim \pi(\cdot|s_t; \theta)$ with $a_t \in \mathcal{A}$, $s_t \in \mathcal{S}$ and $\gamma$ denotes the discounted factor.

## 2.2. Problem Definition

Consistent with previous work (Guo et al., 2023a), we deem a given agent's policy $\pi$ as backdoor-infected if its behaviour follows:

$$\pi(\cdot|s; \theta) = \begin{cases} \pi_{\text{fail}}(s), & \text{if triggered,} \\ \pi_{\text{win}}(s), & \text{otherwise.} \end{cases} \qquad (2)$$

where both $\pi_{\text{win}}$ and $\pi_{\text{fail}}$ take state $s \in \mathbb{R}^{d_s}$ as inputs and generate action $a \in \mathbb{R}^{d_a}$. $\pi_{\text{win}}$ represents a normally well-trained policy which executes following the goal of Equation (1); while $\pi_{\text{fail}}$ represents a policy that aims to cause the victim agent lose the game when the specified trigger explicitly or implicitly appears in the state. For perturbation-based attack (Kiourti et al., 2020), the trigger is directly embedded in the state space, as follows:

$$s_{\text{trg}} = \begin{cases} m \odot s + (1-m) \odot \Delta, & \text{perturbation-based,} \\ s \times a_{\text{trg}} \times a, & \text{adversary-agent.} \end{cases}$$

where $m \in \mathbb{R}^{d_s}, \Delta \in \mathbb{R}^{d_s}$ denote the mask to ensure the trigger location and the corresponding trigger pattern. As for adversary-agent attack (Wang et al., 2021), the state for activating triggers is implicitly generated by complex dynamics between the agents' states and the adversary agent's specific action (*i.e.*, $a_{\text{trg}}$), $s = (s_1, s_2)$ represents the states for the victim and adversary agents, and $a_{\text{trg}}$ and $a$ represent the trigger action sent by the adversary agent and the victim agent's action. In general, $\pi_{\text{fail}}(s)$ is designed to either perform random inappropriate actions (Kiourti et al., 2020) or fail as soon (Wang et al., 2021) as possible to ensure the attack stealthiness and efficacy.

## 2.3. Threat Model

Our threat model contains two parts: *Adversary* and *Defender*. As the Adversary, we assume the adversary can implement arbitrary triggers and manipulate arbitrary episodes to ensure the backdoor efficacy for the victim policy.

Regarding Defender, consistent with previous work (Chen et al., 2023; Guo et al., 2023a), we do not assume access to the adversary training process. Instead, we are provided with a black-box agent embedded with a pretrained policy $\pi$ with its parameters are inaccessible. We are also given a clean environment for testing purposes as previous work (Chen et al., 2023; Guo et al., 2023a; Anonymous, 2025; Guo et al., 2021). This setup simulates a practical scenario where the defender needs to verify an agent's robustness and safety

before deploying it in a potentially poisoned environment. The goal of Defender is to identify whether a given (incoming) state activates a backdoor for the suspicious agent when deploying in a potentially poisoned environment, thereby preventing the execution of backdoored behaviors.

## 2.4. Key Challenges

The test-phase backdoor defense in image classifiers has been well studied (Gao et al., 2019; Guo et al., 2023b; Li et al., 2021), where the trigger behaves in a stateless manner. However, this paper attempts to address test-phase online backdoor defense for a RL policy, which is substantially different and brings new challenges to the research community. On one hand, without accessing the corresponding value network for the suspicious RL policy as well as the parameters, it is difficult to identify the malicious steps for performing backdoors. On the other hand, the defense approach cannot access future steps of a suspicious state within an episode, which imposes an additional and strict constraint on backdoor defense solutions.

## 3. Our Approach: `PolicyGuard`

### 3.1. Overview

`PolicyGuard` is a *test-time* and *step-level* backdoor defense approach for reinforcement learning policies. Given a black-box agent interacting with an environment, `PolicyGuard` aims to identify potentially compromised time steps online by quantifying uncertainty in the agent's behavior using a Gaussian Process (GP) trajectory model.

At a high level, `PolicyGuard` consists of three components: *i),* trajectory encoding that maps state-action pairs into latent step representations through Recurrent Neural Network (Hochreiter & Schmidhuber, 1997); *ii),* Use the collected state-action pairs to fit the parameters within Gaussian Process ; *iii),* uncertainty-based step identification using (adaptive) GP posterior variance. Unlike existing defenses that rely on complete episodes (Yuan et al., 2024), `PolicyGuard` operates without accessing future time steps, allowing for early detection and intervention.

The key insight of `PolicyGuard` is that GP posterior variance naturally quantifies epistemic uncertainty—the model's confidence (Wenger et al., 2022) in its predictions given the observed data. When an agent exhibits backdoor-triggered behaviors, these trajectories differ from the patterns seen during training on episodes collected in the normal environment, resulting in elevated posterior variance.

### 3.2. GP-Based Trajectory Modeling

We adopt an additive GP framework with deep recurrent kernels following (Guo et al., 2021). For each episode $X^{(i)}$

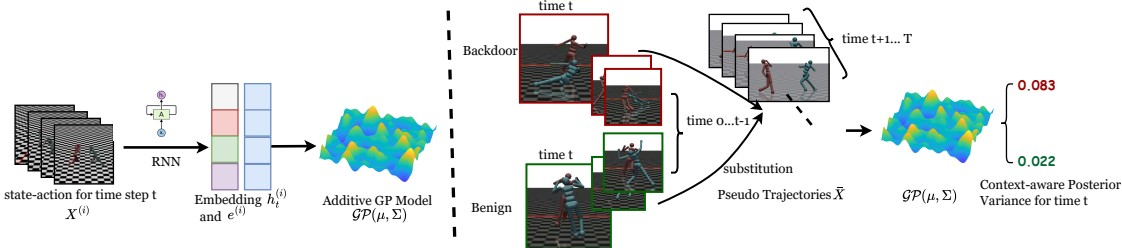

*Figure 1.* **Overview of `PolicyGuard`.** Our approach consists of two phases. In the first phase (*Training GP Model*), we collect state-action pairs across multiple episodes and map them to embedding features $h_t$ using a recurrent neural network. A shallow MLP then projects the final state embedding $h_T^{(i)}$ to $e^{(i)}$ for each episode $X^{(i)}$. These embeddings serve as inputs to a Gaussian Process (GP) model, which is trained to predict the final reward by maximizing the evidence lower bound (ELBO). In the second phase (*Measuring Uncertainty through Adaptive GP Posterior Variance*), for a given suspicious time step t, we construct pseudo trajectories by substituting the state-action pair at collected trajectories with the corresponding historical suspicious state-action pair. These modified trajectories are then passed through the trained GP model to compute posterior variances. We extract and aggregate the variance at time step $t$ across episodes via Interquartile Mean; finally use this uncertainty measure to distinguish backdoor-infected trajectories from benign ones.

within the collected trajectories $\tau$ performed by the target agent, we concatenate state-action pairs $x_t^{(i)} = [s_t^{(i)}, a_t^{(i)}] \in \mathbb{R}^{d_s+d_a}$ and feed them into an LSTM (Hochreiter & Schmidhuber, 1997) encoder $h_\phi$ to obtain latent representations $\{h_t^{(i)}\}_{t=1:T}$ where $d_s$ and $d_a$ represent the dimensions of state and action; $h_t^{(i)} \in \mathbb{R}^q$. We also compute an episodic embedding $e^{(i)} = e_{\phi_1}(h_T^{(i)}) \in \mathbb{R}^q$ through a shallow MLP.

The additive GP model takes the state-action pairs within $N$ episodes $\{X^{(i)}\}_{i=1}^N \in \mathbb{R}^{NT \times (d_s+d_a)}$ as inputs and predicts the final reward, which consists of two components:

$$f = \alpha_t f_t + \alpha_e f_e \qquad (3)$$

where $f_t \sim \text{GP}(0, k_{\gamma_t})$ models timestep correlations via $k_{\gamma_t}(h_t^{(i)}, h_k^{(j)})$, and $f_e \sim \text{GP}(0, k_{\gamma_e})$ captures episode-level patterns via $k_{\gamma_e}(e^{(i)}, e^{(j)})$. Both components use square exponential kernels. For $N$ episodes, the joint prior is:

$$f|X \sim \mathcal{N}(0, k = \alpha_t^2 k_{\gamma_t} + \alpha_e^2 k_{\gamma_e}) \qquad (4)$$

For continuous rewards, we define $y_i|F^{(i)} \sim \mathcal{N}(F^{(i)}w^T, \sigma^2)$ where $F \in \mathbb{R}^{N \times T}$ is the reshaped GP output and $w \in \mathbb{R}^{1 \times T}$ is the mixing weight. For discrete rewards with $K$ classes, we use a linear model activated by softmax with $W \in \mathbb{R}^{K \times T}$. As our additive Gaussian Process (GP) model is trained end-to-end and takes inputs $x \in X$, without loss of generality, we use $x$ to denote the input in the subsequent formulation and optimization of the GP model.

With the purpose of efficient inference, we further define $M$ inducing points $Z = \{z_i\}_{i=1:M}$ in the latent space with corresponding outputs $u$. The variational approximation assumes $q(f, u) = q(u)p(f|u)$ where $q(u) \sim \mathcal{N}(\mu, \Sigma)$. Therefore the conditional prior for GP model is:

$$f|u, X, Z \sim \mathcal{N}(K_{XZ}K_{ZZ}^{-1}u, K_{XX} - K_{XZ}K_{ZZ}^{-1}K_{XZ}^T) \qquad (5)$$

We jointly optimize all model parameters—including the neural encoders ($\Theta_n$), GP kernel parameters ($\Theta_k$), prediction model ($\Theta_p$), inducing point locations ($Z$), and variational parameters ($\{\mu, \Sigma\}$)—by maximizing the evidence lower bound (ELBO), follows below :

$$\log p(y|X, Z, \Theta) \geq \mathbb{E}_{q(f)}[\log p(y|f)] - \text{KL}[q(u)||p(u)]$$

where $\Theta = \{\Theta_n, \Theta_k, \Theta_p\}$ contains encoder, kernel, and prediction parameters. We detail the training procedure and configurations in the Appendix.

### 3.3. Uncertainty Quantification via Adaptive GP Posterior Variance

Once obtaining the GP model, we propose to adapt the GP posterior variance to measure the uncertainty for each time step t, as it captures model confidence and naturally increases for rare or anomalous behaviors. With inducing points $Z$, corresponding outputs $u$ and conditional prior in Equation (5), the variational posterior over $f$ is obtained by marginalizing $q(f) = \int p(f|u)q(u)\,du$, where we have:

$$\mathbb{E}_q[f] = K_{XZ}K_{ZZ}^{-1}\mathbb{E}_{q(u)}[u] = K_{XZ}K_{ZZ}^{-1}\mu \qquad (6)$$
$$\text{Var}_q[f] = \mathbb{E}_{q(u)}[\text{Var}[f|u]] + \text{Var}_{q(u)}[\mathbb{E}[f|u]]$$
$$= K_{XX} - \underbrace{K_{XZ}K_{ZZ}^{-1}K_{ZX}}_{\text{prior variance reduction}} + \underbrace{K_{XZ}K_{ZZ}^{-1}\Sigma K_{ZZ}^{-1}K_{ZX}}_{\text{variational uncertainty}}$$

For the state-action pair $x_t = (s_t, a_t)$ at time $t$, we set the uncertainty score $U(s_t, a_t) = Var_q[f(x_t)]$ as:

$$U(s_t, a_t) = K_{x_t x_t} - K_{x_t Z}K_{ZZ}^{-1}K_{x_t Z}^T \qquad (7)$$
$$+ K_{x_t Z}K_{ZZ}^{-1}\Sigma K_{ZZ}^{-1}K_{x_t Z}^T$$

**Theorem 3.1** (Posterior Variance Bound (Lederer et al., 2019))**.** *Consider a GP with Lipschitz continuous kernel*

$k(\cdot, \cdot)$ *with Lipschitz constant* $L_k$, *observation noise variance* $\sigma^2$. *Let* $\mathcal{B}_\rho(x) = \{x' \in X : \|x' - x_t\| \leq \rho\}$ *denote the time steps restricted to a ball around give time step* $x_t$ *with radius* $\rho$. *Then, for each* $x_t$ *and* $\rho \leq K_{x_t x_t}/L_k$, *the posterior variance is bounded by:*

$$Var_q(x_t) \leq \frac{(4L_k\rho - L_k^2\rho^2)|\mathcal{B}_\rho(x_t)| K_{x_t x_t} + \sigma^2 K_{x_t x_t}}{|\mathcal{B}_\rho(x_t)| (K_{x_t x_t} + 2L_k\rho) + \sigma^2} \tag{8}$$

Theorem 3.1 reveals that the posterior variance for a given $x_t$ is bounded and depends on both the number of neighboring points $|\mathcal{B}_\rho(x_t)|$ and the radius $\rho$. This insight leads to the following Corollary:

**Corollary 3.2.** *Suppose* $x_t \in X$ *or* $x_t$ *is sampled from the same distribution as X, we can have*

$$\lim_{N \to \infty} \rho = 0, \tag{9}$$

$$\lim_{N \to \infty} |\mathcal{B}_\rho(x_t)| = \infty, \tag{10}$$

*then* $\lim_{N \to \infty} Var_q(x_t) = 0$.

Corrolary 3.2 implies that for any benign state-action pair $x_t$, the posterior variance can be driven to zero by collecting sufficient trajectories in $X$.

We propose Theorem 3.1 and Corollary 3.2 to shed light on the discriminability of Gaussian posterior variance between backdoor-infected and benign state-action pairs when GP model is able to learn the manifold of the trajectories on clean environment. The proof as well as the derivation for each equation is detailed in the Appendix.

**Uncertainty Quantification via Pseudo Trajectories.**

As shown in Equation (7), the posterior variance at time step $t$ is computed based on the kernel evaluations involving $x_t$. However, the kernel function $k_\gamma(\cdot, \cdot)$ operates on latent representations produced by the RNN encoder $\phi$, which processes the trajectory sequentially from $t = 1$ to $t = T$. Consequently, the hidden state at time $t$ (*i.e.,*$h_t$) is determined by the entire history up to and including time $t$ (*i.e.,*$h_t = \phi(x_t, h_{t-1})$). Moreover, the episode-level latent representation $e^{(i)}$, which captures global temporal dependencies, is derived from the final hidden state $h_T$. This design implies that meaningful kernel evaluations require access to the *complete* trajectory $X_{1:T}$, not merely depend on the isolated state-action pair $(s_t, a_t)$.

This introduces a key challenge for step-level backdoor detection: at test time, we cannot access the complete state-action sequence $X^{(i)} \in \mathbb{R}^{T \times (d_s + d_a)}$ associated with each suspicious observation in time $t$. Merely considering the current time step's state-action, the posterior variance will become inaccurate. To address this limitation, we introduce `Pseudo Trajectories` that provide surrogate contexts for evaluating each suspicious state-action pair.

Let $\{\tilde{X}^{(i)}\}_{i=1}^N$ denote a set of reference trajectories collected from the test environment, where each trajectory is defined as:

$$\tilde{X}^{(i)} = \{(\tilde{s}_t^{(i)}, \tilde{a}_t^{(i)})\}_{t=1}^T, \quad \tilde{x}_t^{(i)} = [\tilde{s}_t^{(i)}, \tilde{a}_t^{(i)}] \in \mathbb{R}^{d_s + d_a}. \tag{11}$$

Given a suspicious state-action pair $(s_t, a_t)$ at time step $t$, we construct the $i$-th pseudo trajectory $\tilde{X}^{(i)}$ by concatenating the observed history $\{x_1, \ldots, x_t\}$ with the future segments of the reference trajectory $\tilde{X}^{(i)}$:

$$\tilde{X}^{(i)} = \{x_1, \ldots, x_{t-1}, x_t, \tilde{x}_{t+1}^{(i)}, \ldots, \tilde{x}_T^{(i)}\} \tag{12}$$

This construction preserves the valid temporal structure required by the RNN encoder while embedding the suspicious pair at the appropriate time step. Each pseudo trajectory $\tilde{X}^{(i)}$ is then passed through the encoder to obtain the latent representations, from which we compute the posterior variance $\Sigma_t^{(i)}$ following Equation (7).

We aggregate the posterior variances across all pseudo trajectories via Interquartile Mean (IQM) (Wan et al., 2014) and re-define the uncertainty score for $(s_t, a_t)$ as:

$$\mathcal{U}(s_t, a_t) = \text{IQM}\left(\{\Sigma_t^{(i)}\}_{i=1}^N\right). \tag{13}$$

The intuition behind this formulation is that the posterior variance $\Sigma_t^{(i)}$ quantifies the uncertainty of the GP model on state-action pairs that deviate from the training distribution yield higher variance since $K_{x_t Z} K_{ZZ}^{-1} K_{Z x_t}$ becomes small when $x_t$ lies far from the inducing points in the input space. We further propose Theorem 3.3 to prove our intuition.

**Theorem 3.3** (Detection Efficacy)**.** *Let* $U(s_t, a_t)$ *denote the uncertainty score defined in Equation* (13). *With the assumption that for the backdoor-infected state-action pair* $(s_t^{\text{trg}}, a_t^{\text{trg}})$ *and a benign pair* $(s_t, a_t)$ *sampled from the clean environment, then, the expected uncertainty scores satisfy:*

$$\mathbb{E}[U(s_t^{\text{trg}}, a_t^{\text{trg}})] > \mathbb{E}[U(s_t, a_t)]. \tag{14}$$

The proof is detailed in the Appendix.

## 4. Experiment

### 4.1. Experimental Setup

**Environment and Attacks** We follow the settings of previous work (Yuan et al., 2024), we select TrojDRL (Kiourti et al., 2020) as the evaluated *perturbation-based attack* for single-player environments. We attach a 3x3 patch at the left top / right bottom corner as the potential trigger filled with random pixels. We perform TrojDRL in the targeted attack manner to ensure high attack efficacy. The trojan behavior (action) is set randomly while ensuring a high attack efficacy. We follow TrojDRL and select three Atari games - Pong, Breakout, and Space Invaders. Regarding

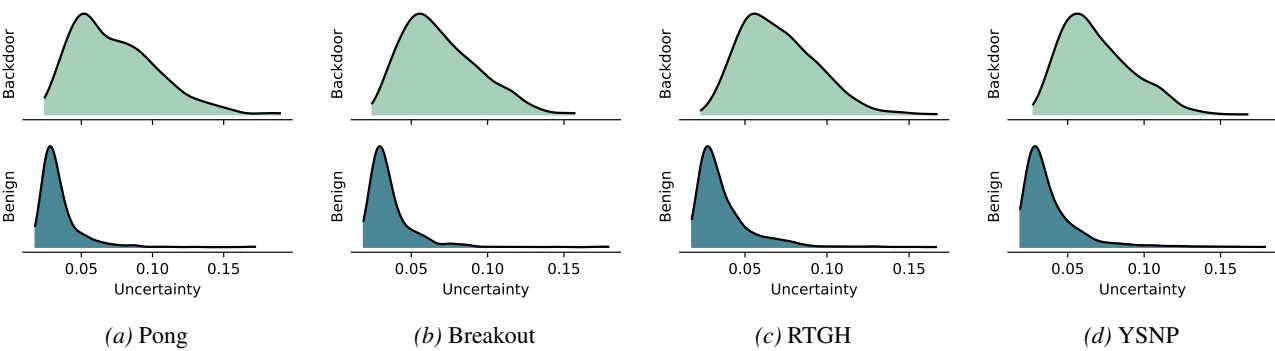

*Figure 2.* Distribution of uncertainty scores (adaptive Gaussian posterior variance) for backdoor and benign time steps across various games. Backdoor attacks use default settings. The distributions for additional games are included in the Appendix.

*Table 1.* Performance (AUROC) on seven widely-studied RL games. Pong, Breakout, and Space Invaders are evaluated under *perturbation-based* attacks, while RTGA, RTGH, YSNP, and Sumo are evaluated under *adversary-agent* attacks. The best result for each game is marked in **boldface**, and *Overall (Avg.) (Perturbation-based / Adversary-agent)* is computed by averaging AUROC scores within each attack category. Note that STRIP and Neural Cleanse require access to predicted probability vectors, whereas other methods rely solely on state-action pairs. B3D (Dong et al., 2021) here is the black-box implementation of NC for adaption to our hard-coded settings. Additionally, SHINE requires the complete episode trajectories for prediction.

| Strategy | Method | Pong | Breakout | Space Invaders | RTGA | RTGH | YSNP | Sumo | Overall (Avg.) |
|---|---|---|---|---|---|---|---|---|---|
| Original | NC | 0.694 | 0.743 | 0.704 | - | - | - | - | 0.714 / ———- |
| | STRIP | **0.879** | **0.904** | **0.917** | - | - | - | - | **0.900** / ———- |
| | SCALE-UP | 0.832 | 0.843 | 0.847 | - | - | - | - | 0.841 / ———- |
| | PolicyCleanse | - | - | - | 0.617 | 0.759 | 0.786 | 0.708 | ———- / 0.718 |
| | SHINE | 0.811 | 0.817 | 0.829 | **0.781** | 0.789 | 0.802 | 0.816 | 0.819 / 0.796 |
| | `PolicyGuard` (Ours) | 0.857 | 0.869 | 0.842 | 0.766 | **0.895** | **0.896** | **0.881** | 0.856 / **0.859** |
| Hard-coded | B3D (Black-box NC) | 0.500 | 0.500 | 0.500 | - | - | - | - | 0.500 / ———- |
| | STRIP | 0.475 | 0.489 | 0.482 | - | - | - | - | 0.482 / ———- |
| | SCALE-UP | 0.495 | 0.462 | 0.471 | - | - | - | - | 0.476 / ———- |
| | PolicyCleanse | - | - | - | 0.502 | 0.502 | 0.506 | 0.503 | ———- / 0.503 |
| | SHINE | 0.817 | 0.821 | 0.841 | 0.803 | 0.796 | 0.816 | 0.826 | 0.826 / 0.810 |
| | `PolicyGuard` (Ours) | **0.861** | **0.878** | **0.864** | **0.827** | **0.904** | **0.901** | **0.881** | **0.868** / **0.878** |

the *adversarial agent attack*, we use the only existing attack (*i.e.,*BackdooRL (Wang et al., 2021)), designed for two-player competitive RL. We evaluate our approach in four games, i.e., You-Shall-NotPass, Sumo-Humans, Run-To-GO-Ants and Run-To-GO-Human. We follow BackdooRL and set the trigger actions as failing immediately.

Notably, since we follow previous work in performing defense under black-box scenarios, we also consider an additional practical attack strategy applicable to both perturbation-based and adversarial agent-based attacks. Inspired by prior work, we implement each attack in a `hard-coded manner`. Specifically, rather than training the policy to learn backdoor behavior, we hard-code the backdoor behavior into a separate adversarial policy and embed it alongside the victim policy. When the adversarial trigger action or perturbation is observed, the adversarial policy is automatically activated. The entire procedure can be formulated as an "*if*-conditional statement". We argue that this type of hard-coded attack is both stealthy and prac-

tical in real-world black-box settings, and can easily bypass existing optimization-based defenses. Implementation details for each attack are provided in the Appendix.

**Configurations for `PolicyGuard`** We select Convolution Neural Network (CNN) (LeCun & Bengio, 1998) as the state (observation) encoder for Pong, Break Out and Space Invader games; while using MLP as the state (observation) encoder for RTGA, RTGH, YSNP, Sumo games. For all games, we use Gated Recurrent Unit (GRU) (Dey & Salem, 2017; Hochreiter & Schmidhuber, 1997) to model each state-action pair for our approaches. We follow previous work to collect trajectories for each environment and set the maximum length of each episode $T$ as 200. We collect 20,000 episodes data to fit our Gaussian Process (GP) model. We implement the GP model with `GPytorch` (Gardner et al., 2018) package and set the training iterations as 200.

**Models and Data Selection** We follow default setting for each type of attack for model selection. Specifically, we use CNN for performing perturbation-based attacks. We

use MLP and LSTM to perform adversary-agent attacks. Regarding the validation data, we evenly and randomly sampled state-action pairs from trajectories performed by benign and backdoor policies. The total amount for the validation data is 10,000 pairs. Notably, all evaluated policies are held-out for the policies used for training GP models.

**Baselines** We follow previous work to adapt several widely-studied test-time backdoor defenses approach into our considered scenarios, such as STRIP, SCALE-UP, NC, B3D (Black-box NC). We also adapt existing defense approaches for RL for comparison, *i.e.,* PolicyCleanse, SHINE. Other backdoor defense approaches (Chen et al., 2023; Bharti et al., 2022; Anonymous, 2025) cannot be applied to our considered scenarios. The details implementation and adaption for each approach are included in the appendix.

**Evaluation Metrics** Similarly to previous work (Guo et al., 2023b), our task is to identify backdoor-infected and benign state-action pair, and we thus adopt the area under receiver operating curve (AUROC) to evaluate all approaches. The larger AUROC implies better detection efficacy.

### 4.2. Experiment Results

We first investigate whether the adaptive GP posterior variance can effectively distinguish backdoor-infected from benign state-action pairs. As shown in Figure 2, the distributions of uncertainty scores exhibit clear separation across all evaluated games. Specifically, backdoor-infected time steps yield higher uncertainty scores than benign ones, validating our core hypothesis that triggered behaviors deviate from the learned behavioral manifold and thus produce elevated posterior variance. This separation is observed for both perturbation-based (*i.e.,* Pong, Breakout) and adversary-agent attacks (*i.e.,* RTGH, YSNP), demonstrating the generalizability of our uncertainty-based detection mechanism.

Table 1 presents the AUROC scores across seven RL games under two attack strategies. For original attacks, `PolicyGuard` achieves the highest overall AUROC for adversary-agent attacks (*i.e.,* 0.859) and competitive performance for perturbation-based attacks (*i.e.,* 0.856), ranking second only to STRIP. Notably, STRIP and Neural Cleanse require access to predicted probability vectors from the policy network, whereas ours operates solely on state-action pairs, making it more practical under black-box settings.

A key finding is that under the hard-coded attack setting, where optimization-based defenses (*i.e.,* NC, STRIP, SCALE-UP, PolicyCleanse) completely fail, yielding AUROC scores near 0.5. This failure occurs because hard-coded backdoors bypass the learned policy parameters entirely, rendering gradient-based or optimization-based trigger recovery ineffective. In contrast, `PolicyGuard` maintains strong detection performance (*i.e.,* 0.868 for

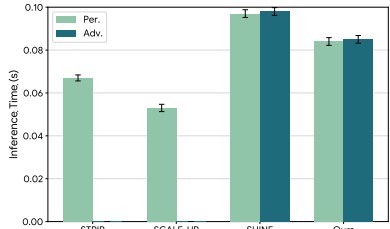

*Figure 3.* The inference latency of various defense approaches. NC and PolicyCleanse are excluded as their computational cost is dominated by model-level reverse engineering, making them inapplicable to step-level test-time latency evaluation.

perturbation-based, 0.878 for adversary-agent attacks), outperforming all baselines including SHINE. These results highlight that trajectory-modeling-based defenses offer superior robustness in rather practical black-box scenarios.

Finally, we evaluate the efficiency of `PolicyGuard` compared with various defense approaches against perturbation-based and adversary-agent attacks. As shown in Figure 3, our approach exhibits slightly higher latency than STRIP and SCALE-UP for perturbation-based attacks, but outperforms SHINE by approximately 9% across all attack types. Notably, STRIP and SCALE-UP are only applicable to perturbation-based attacks under their original design. In contrast, our approach achieves both strong generalizability and competitive efficiency across diverse attack scenarios.

### 4.3. Ablation Study

We further perform various ablation studies (*i.e.,* trigger size, length of trigger, *etc*) to better understand our approach.

**The Effect of Patch Size.** We first investigate the impact of patch size on our approach's detection efficacy for perturbation-based attacks. Across each evaluated environment and backdoor trigger size, we train three different backdoor models to compare their uncertainty with the corresponding benign model's. As shown in Figure 4(a), we observe that our approach performs consistently effective under varying trigger sizes from $3 \times 3$ to $6 \times 6$.

**The Effect of Trigger Action Length.** We evaluate the performance of our approach given varying lengths of trigger actions performed by the adversary agent. The results are shown in Figure 4(b). We observe that our approach performs resilient with trigger actions of varying length from 5 to 20 steps. Specifically, RTGA and YSNP maintain stable AUROC values above 0.85 across all trigger lengths. Such results demonstrate the robustness of our approach against trigger actions with different lengths.

**The Effect of Pseudo Trajectory Size.** We investigate the impact of pseudo trajectory size for approximating uncertainty. As shown in Figure 4(c), we vary the size of pseudo trajectories from 16 to 256. The results show that when the size is larger than 64, our approach achieves optimal perfor-

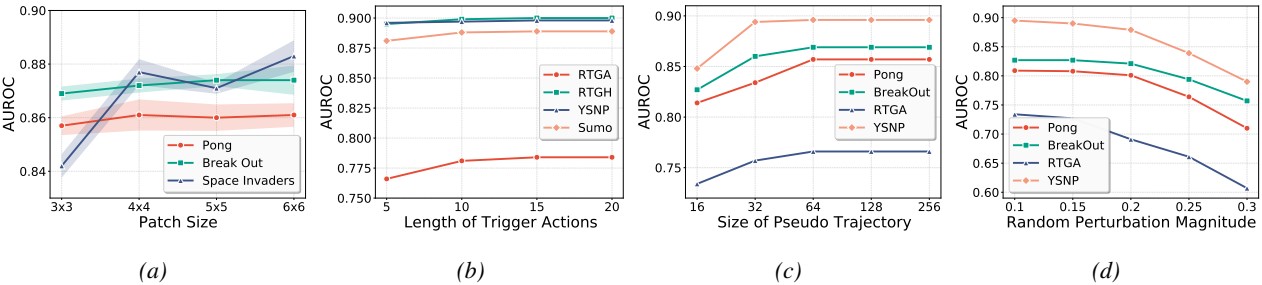

*Figure 4.* Ablation Study. (a) The effect of patch size to our approach's detection efficacy for perturbation-based attacks. Across each evaluated environment and backdoor trigger size, we train three different backdoor models to compare their uncertainty with the corresponding benign model's. (b) The effect of performed trigger actions' length for adversary-agent attacks. (c) The effect of size of pseudo trajectories for approximating uncertainty. (d) The robustness of our approach against random perturbations.

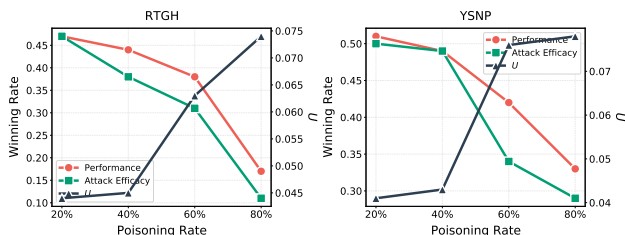

*Figure 5.* The Performance, Attack Efficacy, and Uncertainty $U$ under varying poisoning rates for our considered adaptive attack. *Winning Rate* measures Performance (trigger absent) and Attack Efficacy (trigger present); while $U$ measures the detection efficacy.

mance across all evaluated environments. Pong consistently achieves the highest AUROC, followed by Breakout, RTGA, and YSNP. When the trajectory size is smaller than 64, we observe a noticeable performance degradation, suggesting that sufficient trajectory samples are necessary for reliable uncertainty estimation.

**The Robustness Against Random Perturbations.** We evaluate the robustness of our approach against random perturbations. We design this experiment by injecting each state with random noises of varying magnitudes. Our results (Figure 4 (d)) show that our approach performs effective when the magnitude of random perturbations is smaller than 0.2. When the perturbation magnitude exceeds 0.2, we observe a significant performance drop. Pong and Breakout demonstrate more stable performance, with AUROC only slightly affected when the magnitude approaches 0.2. We speculate that this robustness stems from the architectural differences in state encoders: Pong and Breakout employ CNN-based encoders, whereas RTGA and YSNP utilize MLP-based encoders. CNNs are known to exhibit greater robustness against input perturbations due to their local connectivity and weight sharing properties (Benz et al., 2021).

**The Effectiveness Against the Most Recent Attack** We further evaluate PolicyGuard against Q-Inception (Rathbun et al., 2025), the most recent backdoor attack against RL. As Q-Inception is designed as a perturbation-based attack, we follow the same trigger settings as TrojDRL ($4 \times 4$

*Table 2.* Detection performance of PolicyGuard against Q-Inception on Atari games.

| Metric | Pong | Breakout |
|--------|------|----------|
| AUROC | 0.843 | 0.864 |
| FPR | 0.101 | 0.097 |
| Recall | 0.753 | 0.796 |

random square). We evaluate on two Atari games (Pong and Breakout), training each for ∼40M time steps to ensure optimal attack efficacy. As shown in Table 2, PolicyGuard achieves strong detection performance against Q-Inception, with AUROC above 0.84 and FPR around 0.10 on both games.

**Robustness Against Adaptive Attacks.** We evaluate PolicyGuard against adaptive adversaries aware of our detection mechanism. We consider worst-case scenarios in multi-player environments (*i.e.,*RTGH, YSNP) where attackers optimize trojan actions using the GP model to evade detection. Following the adaptive strategy from (Guo et al., 2021), we adjust malicious trojan actions by adding noise to the victim policy's normal actions until the corresponding uncertainty $U$ matches the benign Interquartile Mean ($U \approx 0.035$). We then select a random action sequence as the trigger and train a trojan agent associating these triggers with the noised trojan actions, keeping other configurations at default settings in BackdooRL (Wang et al., 2021).

Interestingly, the adaptive trojan agent cannot be activated at the default poisoning rate (*i.e.,* 20%) for both environments. We progressively increase the poisoning rate and evaluate the trojan agent's performance (winning rate when triggers are absent) and IQM($U$) under the updated GP model when triggers are present. Results in Figure 5 reveal a tradeoff: when the poisoning rate is $\leq 40\%$, trojan behaviors fail to implant. Once it exceeds 40%, attack efficacy increases but performance drops significantly, and the uncertainty $U$ rises substantially above benign levels. These results demonstrate that PolicyGuard effectively prevents adaptive attacks in

practice, where adversaries cannot simultaneously maintain attack efficacy, clean performance, and stealthiness. More details and ablation studies are included in the Appendix.

## 5. Conclusion

We presented `PolicyGuard`, a test-time, step-level backdoor defense for reinforcement learning agents. By modeling normal agent behavior with an additive Gaussian Process and leveraging posterior variance as an uncertainty signal, `PolicyGuard` can identify backdoor-infected time steps without requiring access to model parameters or complete trajectories. Our theoretical analysis establishes the discriminability of GP posterior variance between benign and backdoored behaviors, while extensive experiments across diverse environments and attack types demonstrate state-of-the-art detection performance, including robustness against hard-coded attacks that defeat existing defenses. These results highlight the effectiveness of uncertainty-aware trajectory modeling for securing RL agents during deployment.

## Impact Statement

This work aims to improve the security and reliability of reinforcement learning systems deployed in safety-critical and adversarial environments. By enabling test-time, step-level detection of backdoor behaviors under black-box assumptions, `PolicyGuard` contributes to mitigating hidden failure modes in RL agents without requiring invasive access to model internals. Potential positive impacts include safer deployment of RL in applications such as autonomous control, robotics, and multi-agent systems. As with any defensive technology, there is a risk that insights into detection mechanisms could inform adaptive attacks; however, our approach relies on general uncertainty principles rather than trigger-specific heuristics, reducing misuse potential. Overall, we believe the benefits of strengthening RL robustness outweigh the associated risks.

## Acknowledgements

This work was partially supported by NSF IIS 2347592, 2348169, DBI 2405416, CCF 2348306, CNS 2347617, RISE 2536663.

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

## A. Limitations

**Clean Environment Assumption.** PolicyGuard requires access to a clean environment for collecting reference trajectories and training the GP model. While this assumption is standard across existing RL backdoor defenses (PolicyCleanse, SHINE, BIRD), it may not always be feasible in real-world scenarios where the deployment environment differs significantly from the test environment. Bridging this gap through domain adaptation or sim-to-real transfer techniques is an important direction for future work.

**Detection Only.** PolicyGuard focuses on detecting backdoor-infected steps and does not include a built-in intervention or mitigation mechanism. Upon detection, the system can be composed with existing input-level mitigation approaches ( SHINE) or simple intervention strategies such as executing safe default actions. Developing a unified detect-intervene-mitigate pipeline remains future work.

**Gap Between Simulation and Real World.** Our evaluation is conducted entirely in simulated environments (Atari, MuJoCo). Real-world RL deployments ( robotics, autonomous driving) involve noisy sensor inputs, partial observability, non-stationary dynamics, and physical constraints that are not fully captured by simulators. Whether the GP posterior variance remains a reliable uncertainty signal under these real-world conditions has not been validated. Developing robust uncertainty quantification methods that maintain detection efficacy across the sim-to-real transfer gap is a critical challenge for deploying PolicyGuard in practice.

## B. Code

The codes are included in the Supplementary File.

## C. Detailed Procedure for Building GP Model

In this section, we provide a comprehensive overview of the self-explainable Gaussian Process (GP) model construction, detailing the feature extraction process and the additive GP framework with deep recurrent kernels. This procedure follows the methodology established in EDGE (Guo et al., 2021).

### C.1. Feature Extraction and Trajectory Encoding

To capture the temporal dependencies within reinforcement learning episodes, we utilize a deep neural network architecture to map raw state-action pairs into latent representations.

**State Encoding** For high-dimensional inputs such as images (e.g., Atari games), we employ a Convolutional Neural Network (CNN) as the encoder. The CNN processes the raw pixel state $s_t$ at time step $t$ to extract spatial features. For environments with low-dimensional states (e.g., MuJoCo), a Multi-Layer Perceptron (MLP) is used.

$$z_t = \texttt{Encoder}(s_t; \theta_{enc})$$

**Sequential Encoding** The state embeddings $z_t$ are concatenated with the corresponding actions $a_t$ and fed into a Gated Recurrent Unit (GRU) (Hochreiter & Schmidhuber, 1997; Dey & Salem, 2017) to model the sequential dynamics of the episode. The GRU maintains a hidden state $h_t$ that summarizes the history up to time $t$:

$$h_t = \texttt{GRU}([z_t, a_t], h_{t-1}; \theta_{rnn})$$

The final hidden state of the episode, $h_T$, serves as a compact summary of the entire trajectory. To obtain the episode-level embedding $e^{(i)}$, we pass $h_T$ through a projection MLP:

$$e^{(i)} = \texttt{MLP}(h_T^{(i)}; \phi)$$

### C.2. Additive Gaussian Process with Deep Recurrent Kernels

We model the correlation between episodes and predict the final reward using an additive Gaussian Process. Let $f$ be the function mapping episode embeddings to rewards. We assume $f$ follows a GP prior with a zero mean and a kernel function $k(\cdot, \cdot)$.

To capture distinct features of the policy's behavior, we employ an additive kernel formulation. The function $f$ is decomposed into a weighted sum of $J$ independent GPs, i.e., $f(x) = \sum_{j=1}^{J} \alpha_j f_j(x)$, where $\alpha_j$ are scalar weights. The resulting kernel is:

$$k(e^{(i)}, e^{(k)}) = \alpha_1^2 k_t + \alpha_2^2 k_e$$

where each kernel $k_t$ and $k_t$ represent the kernels for steps and episodes.

## D. Derivation of the Evidence Lower Bound (ELBO)

To efficiently train the GP model and the neural networks end-to-end, we employ Variational Inference (VI) with inducing points. Here, we detail the derivation of the Evidence Lower Bound (ELBO) used as the maximization objective.

### D.1. Variational Distribution

Direct inference in GPs scales cubically with the number of data points $N$ (i.e., $\mathcal{O}(N^3)$). To address this, we introduce a set of $M$ inducing points $Z = \{z_m\}_{m=1}^{M}$ and their corresponding function values $u = f(Z)$. We define a variational distribution $q(f)$ to approximate the true posterior $p(f|y)$. Following the sparse GP framework, we assume $q(u)$ is a multivariate Gaussian:

$$q(u) = \mathcal{N}(u|m, S)$$

where $m \in \mathbb{R}^M$ and $S \in \mathbb{R}^{M \times M}$ are variational parameters learned during training. The conditional distribution $p(f|u)$ is fixed by the GP prior, leading to the joint variational distribution:

$$q(f) = \int p(f|u)q(u)du$$

## E. Technical Details for ELBO Derivation

In this section, we provide the comprehensive derivation of the Evidence Lower Bound (ELBO) used in our method, corresponding to the optimization objective discussed in Section 3.3 of the main paper. We also detail the "whitening" operation and the analytical form of the expected conditional log-likelihood.

### E.1. Derivation of the Evidence Lower Bound

We aim to maximize the marginal likelihood $\log p(y|X, Z)$. By introducing the latent function values $f$ and inducing variables $u$, we derive the ELBO as follows:

$$
\begin{aligned}
\log p(y|X, Z) &= \log \iint p(y, f, u|X, Z) \, du \, df \\
&= \log \iint p(y|f, X)p(f, u|X, Z)\frac{q(f, u)}{q(f, u)} \, du \, df \\
&= \log \mathbb{E}_{q(f,u)}\left[\frac{p(y|f, X)p(f, u|X, Z)}{q(f, u)}\right]
\end{aligned}
\tag{15}
$$

Applying Jensen's inequality, we obtain the lower bound (McShane, 1937):

$$
\begin{aligned}
\log p(y|X, Z) &\geq \mathbb{E}_{q(f,u)}\left[\log p(y|f, X) + \log \frac{p(f, u|X, Z)}{q(f, u)}\right] \\
&= \mathbb{E}_{q(f,u)}[\log p(y|f)] - \mathbb{E}_{q(f,u)}\left[\log \frac{q(f, u)}{p(f, u|X, Z)}\right]
\end{aligned}
\tag{16}
$$

By factorizing the joint distributions $p(f, u|X, Z) = p(f|u)p(u)$ and $q(f, u) = p(f|u)q(u)$, the second term simplifies to the Kullback-Leibler (KL) divergence between the variational and prior distributions of the inducing variables (Gardner et al., 2018):

$$\mathcal{L}_{\text{ELBO}} = \mathbb{E}_{q(f)}[\log p(y|f)] - \text{KL}[q(u)||p(u)] \tag{17}$$

Maximizing this ELBO is equivalent to minimizing the KL divergence between the variational joint distribution $q(f, u)$ and the true posterior $p(f, u|y)$.

# F. Experimental Settings and Configurations

In this section, we detail the architecture configurations and hyperparameters used for the experiments on Atari and MuJoCo environments.

## F.1. Atari and Mujoco Games Configuration

For the high-dimensional visual inputs of Atari games (Mnih, 2013) (e.g., Pong, Breakout), we employ a Convolutional Neural Network (CNN) as the encoder.

- **Input Preprocessing**: The raw game frames are preprocessed into $84 \times 84$ grayscale images. We stack 4 consecutive frames as the state input $s_t$.

- **Encoder Architecture**:
    - Layer 1: $8 \times 8$ convolution with 32 filters, stride 4, followed by ReLU.
    - Layer 2: $4 \times 4$ convolution with 64 filters, stride 2, followed by ReLU.
    - Layer 3: $3 \times 3$ convolution with 64 filters, stride 1, followed by ReLU.
    - The output is flattened and passed through a fully connected layer to produce the state embedding $z_t$.

- **Sequence Modeling**: A Gated Recurrent Unit (GRU) processes the sequence of state embeddings. We typically set the hidden dimension size to 256.

## F.2. MuJoCo Games Configuration

For MuJoCo continuous control tasks (e.g., Run-to-Goal, You-Shall-Not-Pass), the inputs are low-dimensional state vectors.

- **Encoder Architecture**: We use a Multi-Layer Perceptron (MLP) encoder.
    - Two fully connected layers with 64 units each.
    - ReLU activation functions are applied after each layer.

- **Sequence Modeling**: A GRU with a hidden dimension of 128 is used to capture the temporal dependencies of the trajectory.

## F.3. Optimization and Hyperparameters

The model is trained end-to-end by maximizing the ELBO.

- **Inducing Points**: We use $M = 600$ inducing points for the sparse Gaussian Process to balance computational efficiency and approximation capability.

- **Optimizer**: We utilize the Adam optimizer.

- **Learning Rates**:
    - GP Parameters (variational parameters, kernel hyperparameters): $1 \times 10^{-3}$.
    - Neural Network Parameters (Encoder, GRU): $1 \times 10^{-4}$.

- **Batch Size**: Training is performed with a batch size of 64 episodes.

# G. Proof and Derivation

### G.1. Derivation of Efficient Posterior Variance

We derive the variational posterior in Equation (7) from the sparse GP conditional prior in Equation (5).

*Table 3.* Architecture and Hyperparameter Configuration for Atari Games (e.g., Pong, Breakout).

| Parameter | Value / Configuration |
|---|---|
| *Input Preprocessing* | |
| Input Dimensions | $84 \times 84$ (Grayscale) |
| Frame Stacking | 4 consecutive frames |
| *CNN Encoder Architecture* | |
| Layer 1 | 32 filters, $8 \times 8$ kernel, stride 4, ReLU |
| Layer 2 | 64 filters, $4 \times 4$ kernel, stride 2, ReLU |
| Layer 3 | 64 filters, $3 \times 3$ kernel, stride 1, ReLU |
| Output | Flattened vector |
| *Sequential Modeling* | |
| Recurrent Unit | Gated Recurrent Unit (GRU) |
| Hidden Dimension | 256 |
| *Optimization & GP Settings* | |
| Inducing Points ($M$) | 100 |
| Batch Size | 64 |
| Optimizer | Adam |
| Learning Rate (GP) | $1 \times 10^{-3}$ |
| Learning Rate (NN) | $1 \times 10^{-4}$ |

**Sparse GP Prior.** With inducing points $Z$ and outputs $u$, the conditional prior is:

$$f|u, X, Z \sim \mathcal{N}\left(K_{XZ}K_{ZZ}^{-1}u, \; K_{XX} - K_{XZ}K_{ZZ}^{-1}K_{ZX}\right) \tag{18}$$

**Variational Approximation.** We approximate $p(u|y)$ with $q(u) = \mathcal{N}(\mu, \Sigma)$, where $\mu \in \mathbb{R}^M$ and $\Sigma \in \mathbb{R}^{M \times M}$ are learned parameters. The variational posterior over $f$ is obtained by marginalizing:

$$q(f) = \int p(f|u)q(u) \, du \tag{19}$$

**Posterior Mean.** Since $\mathbb{E}[f|u] = K_{XZ}K_{ZZ}^{-1}u$:

$$\mathbb{E}_q[f] = K_{XZ}K_{ZZ}^{-1}\mathbb{E}_{q(u)}[u] = K_{XZ}K_{ZZ}^{-1}\mu \tag{20}$$

**Posterior Variance.** Applying the law of total variance:

$$\begin{aligned}
\text{Var}_q[f] &= \mathbb{E}_{q(u)}[\text{Var}[f|u]] + \text{Var}_{q(u)}[\mathbb{E}[f|u]] \\
&= \underbrace{K_{XX} - K_{XZ}K_{ZZ}^{-1}K_{ZX}}_{\text{prior variance reduction}} + \underbrace{K_{XZ}K_{ZZ}^{-1}\Sigma K_{ZZ}^{-1}K_{ZX}}_{\text{variational uncertainty}}
\end{aligned} \tag{21}$$

For observations $X_{1:t}$ up to timestep $t$, this yields Equation (10):

$$\begin{aligned}
\mu_{1:t}^{(i)} &= K_{X_{1:t}Z}K_{ZZ}^{-1}u \\
\Sigma_{1:t}^{(i)} &= K_{X_{1:t}X_{1:t}} - K_{X_{1:t}Z}K_{ZZ}^{-1}K_{X_{1:t}Z}^T + K_{X_{1:t}Z}K_{ZZ}^{-1}\Sigma K_{ZZ}^{-1}K_{X_{1:t}Z}^T
\end{aligned} \tag{22}$$

The computational complexity reduces from $O((NT)^3)$ to $O(tM^2)$, enabling efficient step-level variance computation.

*Table 4.* Architecture and Hyperparameter Configuration for MuJoCo Games (Bansal et al., 2017) (e.g.,Run-to-Goal, You-shall-Not-Pass).

| Parameter | Value / Configuration |
|---|---|
| *Input Processing* | |
| Input Type | Low-dimensional physical state vector |
| *MLP Encoder Architecture* | |
| Structure | 2 Fully Connected Layers |
| Hidden Units | 64 units per layer |
| Activation | ReLU |
| *Sequential Modeling* | |
| Recurrent Unit | Gated Recurrent Unit (GRU) |
| Hidden Dimension | 128 |
| *Optimization & GP Settings* | |
| Inducing Points ($M$) | 600 |
| Batch Size | 64 |
| Optimizer | Adam |
| Learning Rate (GP) | $1 \times 10^{-3}$ |
| Learning Rate (NN) | $1 \times 10^{-4}$ |

## G.2. Proof for Theorem 3.1

**Theorem G.1.** *Consider a GP with Lipschitz continuous kernel $k(\cdot, \cdot)$ with Lipschitz constant $L_k$, observation noise variance $\sigma^2$. Let $\mathcal{B}_\rho(x_t) = \{x' \in \mathcal{X} : \|x' - x_t\| \leq \rho\}$ denote the time steps restricted to a ball around given time step $x_t$ with radius $\rho$. Then, for each $x_t$ and $\rho \leq K_{x_t x_t}/L_k$, the posterior variance is bounded by*

$$\text{Var}_q(x_t) \leq \frac{(4L_k\rho - L_k^2\rho^2)|\mathcal{B}_\rho(x_t)| \, K_{x_t x_t} + \sigma^2 K_{x_t x_t}}{|\mathcal{B}_\rho(x_t)| \, (K_{x_t x_t} + 2L_k\rho) + \sigma^2} \tag{23}$$

*Proof.* Since $K + \sigma^2 I$ is a positive definite matrix, we have

$$\text{Var}_q(x_t) \leq K_{x_t x_t} - \frac{\|K_{x_t X}\|^2}{\lambda_{\max}(K) + \sigma^2}, \tag{24}$$

where $\lambda_{\max}(K)$ denotes the maximum eigenvalue of $K$.

Applying the Gershgorin circle theorem (Gershgorin, 1931), the maximum eigenvalue is bounded by

$$\lambda_{\max}(K) \leq N \max_{x',x'' \in \mathcal{X}} k(x', x''). \tag{25}$$

Furthermore, by definition of $K_{x_t \mathcal{X}}$, we have

$$\|K_{x_t \mathcal{X}}\|^2 \geq N \min_{x' \in \mathcal{X}} k^2(x', x_t). \tag{26}$$

Therefore, $\text{Var}_q(x_t)$ can be bounded by

$$\text{Var}_q(x_t) \leq K_{x_t x_t} - \frac{N \min_{x' \in \mathcal{X}} k^2(x', x_t)}{N \max_{x',x'' \in \mathcal{X}} k(x', x'') + \sigma^2}. \tag{27}$$

This bound can be further simplified by exploiting the fact that $\text{Var}_q(x_t)$ cannot increase when adding more training samples, and considering only samples inside the ball $\mathcal{B}_\rho(x_t)$ with radius $\rho \in \mathbb{R}_+$. Using this reduced data set instead of $\mathcal{X}$ and

writing the right side as a single fraction results in

$$\text{Var}_q(x_t) \leq \frac{K_{x_t x_t} \sigma^2 + |\mathcal{B}_\rho(x_t)| \, \xi(x_t, \rho)}{|\mathcal{B}_\rho(x_t)| \max_{x', x'' \in \mathcal{B}_\rho(x_t)} k(x', x'') + \sigma^2}, \tag{28}$$

where

$$\xi(x_t, \rho) = K_{x_t x_t} \max_{x', x'' \in \mathcal{B}_\rho(x_t)} k(x', x'') - \min_{x' \in \mathcal{B}_\rho(x_t)} k^2(x', x_t). \tag{29}$$

Under the assumption that $\rho \leq K_{x_t x_t}/L_k$, it follows from the Lipschitz continuity of $k(\cdot, \cdot)$ that

$$\min_{x' \in \mathcal{B}_\rho(x_t)} k^2(x', x_t) \geq (K_{x_t x_t} - L_k \rho)^2. \tag{30}$$

Furthermore, it holds that

$$\max_{x', x'' \in \mathcal{B}_\rho(x_t)} k(x', x'') \leq K_{x_t x_t} + 2L_k \rho. \tag{31}$$

Therefore, $\xi(x_t, \rho)$ can be bounded by

$$\xi(x_t, \rho) \leq K_{x_t x_t}(K_{x_t x_t} + 2L_k \rho) - (K_{x_t x_t} - L_k \rho)^2 \tag{32}$$

$$= K_{x_t x_t}^2 + 2K_{x_t x_t} L_k \rho - K_{x_t x_t}^2 + 2K_{x_t x_t} L_k \rho - L_k^2 \rho^2 \tag{33}$$

$$= 4K_{x_t x_t} L_k \rho - L_k^2 \rho^2. \tag{34}$$

Substituting this bound into the variance inequality yields

$$\text{Var}_q(x_t) \leq \frac{K_{x_t x_t} \sigma^2 + |\mathcal{B}_\rho(x_t)|(4K_{x_t x_t} L_k \rho - L_k^2 \rho^2)}{|\mathcal{B}_\rho(x_t)|(K_{x_t x_t} + 2L_k \rho) + \sigma^2}. \tag{35}$$

Factoring out $K_{x_t x_t}$ from the numerator:

$$\text{Var}_q(x_t) \leq \frac{(4L_k \rho - L_k^2 \rho^2)|\mathcal{B}_\rho(x_t)| \, K_{x_t x_t} + \sigma^2 K_{x_t x_t}}{|\mathcal{B}_\rho(x_t)|(K_{x_t x_t} + 2L_k \rho) + \sigma^2}. \tag{36}$$

$\square$

### G.3. Proof for Theorem 3.3

**Theorem G.2** (Detection Consistency). *Let $U(s_t, a_t)$ denote the uncertainty score defined in Equation (13). For a backdoor-infected state-action pair $(s_t^{\text{trg}}, a_t^{\text{trg}})$ and a benign pair $(s_t, a_t)$ sampled from the clean behavioral distribution, assume the following conditions hold:*

1. *The GP model is trained on trajectories collected from a clean environment.*

2. *The inducing points $Z = \{z_i\}_{i=1}^M$ are learned to represent the clean behavioral manifold.*

3. *The backdoor-triggered latent representation satisfies $\min_{z_i \in Z} \|h_t^{\text{trg}} - z_i\| > \min_{z_i \in Z} \|h_t - z_i\| + \delta$ for some $\delta > 0$.*

*Then, the expected uncertainty scores satisfy:*

$$\mathbb{E}[U(s_t^{\text{trg}}, a_t^{\text{trg}})] > \mathbb{E}[U(s_t, a_t)]. \tag{37}$$

*Proof.* From Equation (7), the posterior variance at time step $t$ for the $i$-th pseudo trajectory is:

$$\Sigma_{1:t}^{(i)} = K_{X_{1:t} X_{1:t}} - K_{X_{1:t} Z} K_{ZZ}^{-1} K_{X_{1:t} Z}^\top + K_{X_{1:t} Z} K_{ZZ}^{-1} \Sigma K_{ZZ}^{-1} K_{X_{1:t} Z}^\top. \tag{38}$$

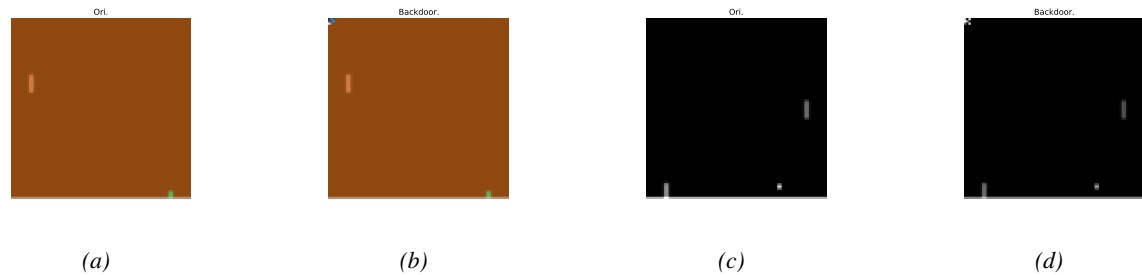

|        |        |        |        |
| :----: | :----: | :----: | :----: |
| *(a)*  | *(b)*  | *(c)*  | *(d)*  |

*Figure 6.* The Benign and Backdoor-infected Frame for Pong Game. The left pair (a-b) represents the original colorful frames, while the right pair (c-d) represents the (pre-processed) gray-scale frames.

For the squared exponential (SE) kernel $k_\gamma(h, h') = \exp\left(-\frac{\|h-h'\|^2}{2\gamma^2}\right)$, the diagonal entry of the posterior variance at time $t$ can be decomposed as:

$$\Sigma_t^{(i)} = k_\gamma(h_t, h_t) - \mathbf{k}_t^\top K_{ZZ}^{-1}\mathbf{k}_t + \mathbf{k}_t^\top K_{ZZ}^{-1}\Sigma K_{ZZ}^{-1}\mathbf{k}_t, \tag{39}$$

where $\mathbf{k}_t = [k_\gamma(h_t, z_1), \ldots, k_\gamma(h_t, z_M)]^\top \in \mathbb{R}^M$ is the kernel vector between the latent representation $h_t$ and the inducing points.

Note that $k_\gamma(h_t, h_t) = 1$ for any $h_t$. The term $\mathbf{k}_t^\top K_{ZZ}^{-1}\mathbf{k}_t$ represents the variance reduction from the inducing points, which is larger when $h_t$ is closer to the inducing points.

For a backdoor-infected pair, the RNN encoder produces a latent representation $h_t^{\text{trg}} = \phi(x_t^{\text{trg}}, h_{t-1})$ that encodes the anomalous state-action pair. By Assumption 3, this representation lies farther from all inducing points:

$$\|h_t^{\text{trg}} - z_i\| > \|h_t - z_i\| + \delta, \quad \forall z_i \in Z. \tag{40}$$

Since the SE kernel is monotonically decreasing in distance, we have:

$$k_\gamma(h_t^{\text{trg}}, z_i) = \exp\left(-\frac{\|h_t^{\text{trg}} - z_i\|^2}{2\gamma^2}\right) < \exp\left(-\frac{(\|h_t - z_i\| + \delta)^2}{2\gamma^2}\right) < k_\gamma(h_t, z_i). \tag{41}$$

Let $\mathbf{k}_t^{\text{trg}}$ and $\mathbf{k}_t$ denote the kernel vectors for the backdoor-infected and benign cases, respectively. The element-wise inequality $\mathbf{k}_t^{\text{trg}} < \mathbf{k}_t$ implies:

$$(\mathbf{k}_t^{\text{trg}})^\top K_{ZZ}^{-1}\mathbf{k}_t^{\text{trg}} < \mathbf{k}_t^\top K_{ZZ}^{-1}\mathbf{k}_t, \tag{42}$$

since $K_{ZZ}^{-1}$ is positive definite (as $K_{ZZ}$ is a valid kernel matrix).

Consequently, the variance reduction term is smaller for backdoor-infected pairs, yielding higher posterior variance:

$$\Sigma_t^{(i),\text{trg}} = 1 - (\mathbf{k}_t^{\text{trg}})^\top K_{ZZ}^{-1}\mathbf{k}_t^{\text{trg}} + (\text{variational term}) > \Sigma_t^{(i)}. \tag{43}$$

Since the IQM aggregation in Equation (13) preserves ordering (as a weighted average of order statistics), we conclude:

$$U(s_t^{\text{trg}}, a_t^{\text{trg}}) = \text{IQM}\{\Sigma_t^{(i),\text{trg}}\}_{i=1}^N > \text{IQM}\{\Sigma_t^{(i)}\}_{i=1}^N = U(s_t, a_t). \tag{44}$$

$\square$

# H. Experimental Setup

## H.1. Evaluated Games and Agents.

We evaluate PolicyGuard on three Atari games (Mnih, 2013)—Pong, Breakout, and SpaceInvaders and four MuJoCo two-player competitive tasks (Bansal et al., 2017)—Run-To-Goal-Ants, Run-To-Goal-Human, You-Shall-Not-Pass and Sumo.

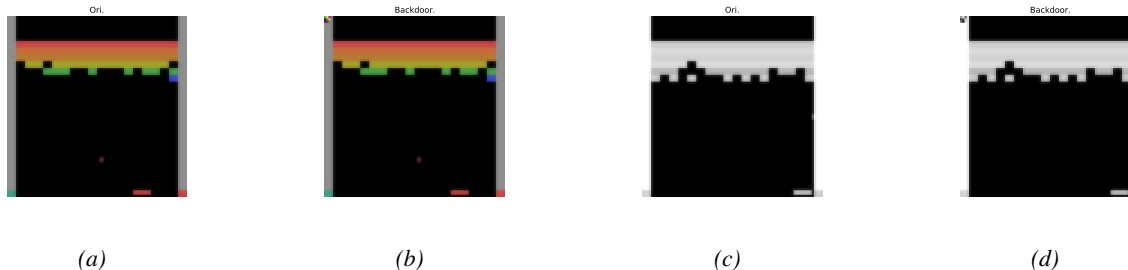

*(a)*       *(b)*       *(c)*       *(d)*

*Figure 7.* The Benign and Backdoor-infected Frame for Breakout Game. The left pair (a-b) represents the colorful frames, while the right pair (c-d) represents the (pre-processed) gray-scale frames.

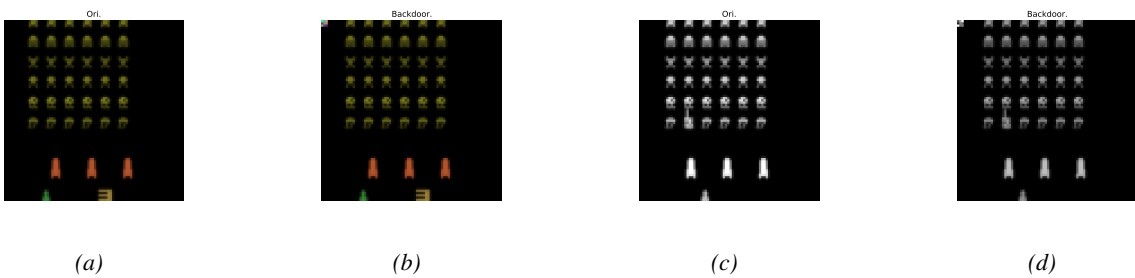

*(a)*       *(b)*       *(c)*       *(d)*

*Figure 8.* The Benign and Backdoor-infected Frame for SpaceInvaders Game. The left pair (a-b) represents the colorful frames, while the right pair (c-d) represents the (pre-processed) gray-scale frames.

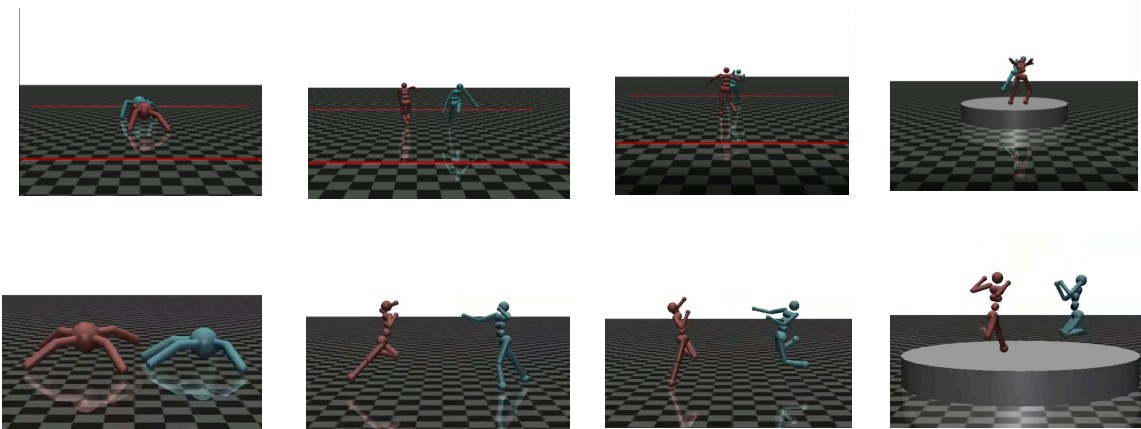

*Figure 9.* **The visual demonstration of trigger actions performed by the adversary agent.** The top row represent the benign agent's behavior. From left to right, each game is Run-to-Goal-Ants, Run-to-Goal-Humans, You-Shall-Not-Pass, Sumo. The bottom row represents the corresponding trigger actions performed the trigger agents (Blue agent for Ants, Red agent for the humans.) The trigger actions are randomly sampled within 8- and 17- dimension spaces for ant and humans, respectively.

For each environment, we obtain benign agents from official pre-trained model repositories (Atari: https://github.com/greydanus/baby-a3c; MuJoCo: https://github.com/openai/multiagent-competition) and construct corresponding backdoor-infected agents following the attack configurations in prior work (Kiourti et al., 2020; Wang et al., 2021). Specifically, the backdoor triggers and poisoning strategies adhere to the threat models established in these studies, ensuring our evaluation covers realistic attack scenarios. All agents are trained using policy gradient methods (PPO for Atari games and SAC/PPO for MuJoCo tasks). For each game, we collect trajectory datasets by rolling out both benign and backdoor-infected policies across multiple random seeds, yielding training and testing splits for our detection experiments.

For backdoor-infected trajectories, we only collect the first four time steps after trigger activation. This design choice reflects our empirical observation that backdoor-infected agents typically require $\geq 7$ (Atari) or 10 (MuJoCo) consecutive malicious steps to cause task failure. By focusing on early-stage detection, we simulate realistic deployment scenarios

where PolicyGuard can intervene and prevent catastrophic outcomes before the backdoor behavior fully manifests. And for each game, we follow previous work (Wang et al., 2021) to set $p = 0.2$ for each time step to attach the backdoor pattern or perform trigger actions, once the time step initializes the backdoor behavior, $p$ is set as 100% for the the remaining time steps until the game ends.

**Hard-coded Attack Settings** Different from attacks (Kiourti et al., 2020; Wang et al., 2021) that embed adversarial and benign behaviors into a single learned policy, hard-coded attacks manipulate the victim agent to perform malicious behavior through the following mechanism:

$$\pi_{\text{backdoor}}(s) = \begin{cases} \pi_{\text{fail}}(s), & \text{if trigger pattern/action is present} \\ \pi_{\text{clean}}(s), & \text{otherwise} \end{cases} \tag{45}$$

We argue that such hard-coded attacks can be easily and effectively deployed in black-box settings. When the victim agent observes the trigger pattern or action, it automatically executes the failure policy and misbehaves. Crucially, this entire process does not involve any learning or optimization, making gradient-based and optimization-based defenses completely ineffective.

**Visualization of Trigger Patterns (Actions)** We here visualize the trigger patterns used for perturbation-based attacks for Atari Games (Mnih, 2013) and trigger actions used for Mujoco Games (Bansal et al., 2017). We show the visualization results for Atari (Mnih, 2013) in Figures 6 to 8; while those for Mujoco are Figure 9.

## H.2. The Detailed Implementations for Different Baselines

Notably, for all baselines we evaluate them under our considered clean environment, which means that each approach cannot access the backdoor signals.

- **Neural Cleanse (NC) (Wang et al., 2019)**:

  We use NC to do reverse engineering to recover potential trigger patterns, while use the trigger patterns to tag the backdoor state through matching the neuron activation similarity, which is proposed by NC. For hard-coded settings, we implement B3D (Dong et al., 2021), which is a black-box variants for NC.

- **STRIP (Gao et al., 2019)**:

  We directly follow `https://github.com/garrisongys/STRIP.git` to implement STRIP, and assume that the defender can access the logits of the agents' actions.

- **SCALE-UP (Guo et al., 2023b)**:

  We follow directly `https://github.com/JunfengGo/SCALE-UP.git` to implement SCALE-UP.

- **PolicyCleanse (Guo et al., 2023a)**

  We follow `https://github.com/listentomi/RL-backdoor-detection.git` to implement Policy-Cleanse.

- **SHINE (Yuan et al., 2024)**:

  We manipulate SHINE according to `https://github.com/eurekayuan/SHINE.git` and `https://github.com/Henrygwb/edge.git`. We follow its configurations and allow it to collect the entire episodes' steps and make predictions. We use $u$ as the predictor for determining the pairs of backdoor / benign state-action, to ensure theirs achieve the optimal performance.

## H.3. Distributions for GP Variance across Different Games

The distributions are shown in Figure 10.

## H.4. The Effect of Training Trajectory Size

We here investigate the effect of training data size. The results are shown in Figure 11. We found that using 20K can better balance the training cost and detection efficacy.

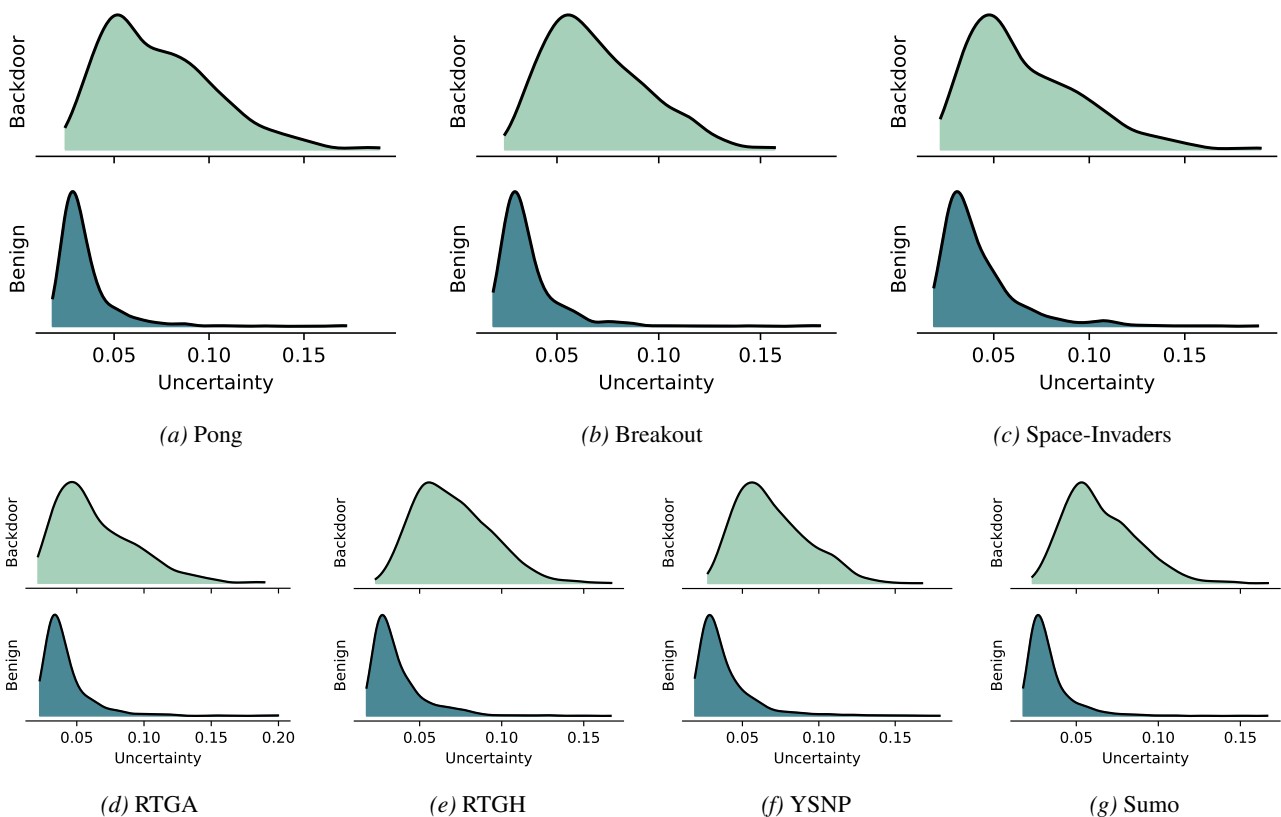

*Figure 10.* The distributions for Posterior Gaussian Variance across different games

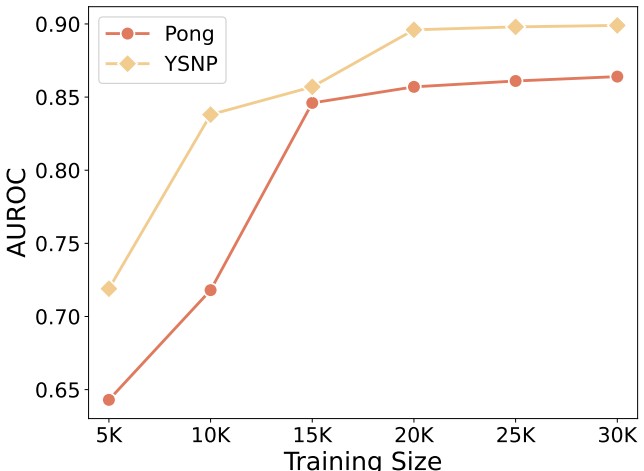

*Figure 11.* The Effect of Training Data Size

## H.5. A Closer Look at the Effectiveness of `PolicyGuard`

We here follow (Guo et al., 2023a) to draw the distributions of benign, backdoor and inducing points Z's hidden features. We can see that in the tSNE clustering approach, benign and inducing Z points can be easily separated with backdoor points.

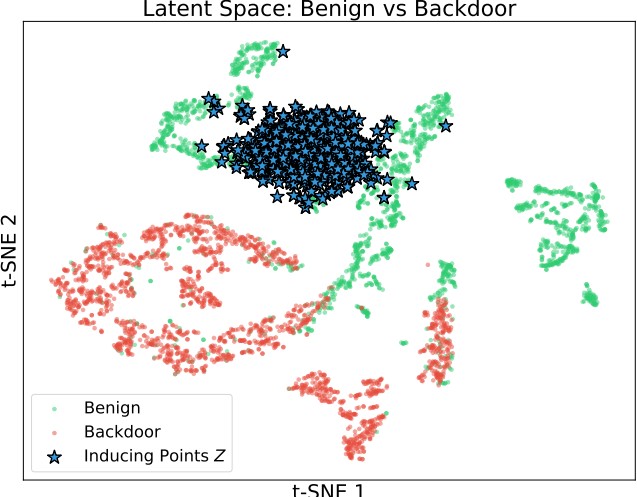

*Figure 12.* The tSNE clusters for benign, backdoor and Z points.

### H.6. Adaptive Perturbation-based Attacks

We also consider an adaptive attack for Perturbation-based Attack, we perform adaptive attack by fixing a random action as the target action and perform reverse engineering as NC (Wang et al., 2019), to optimize a trigger pattern which makes a pre-trained GP variance minimum, as follows:

$$\Delta = \mathbb{E}_{x \sim X}[\arg\min_{\Delta} \Sigma(x \odot m + m \odot \Delta)]. \quad \textit{s.t.} \text{ Preserve the attack efficacy \& Performance} \tag{46}$$

Notably, we randomly initialize the trigger pattern during the optimization process. We use the optimized trigger patterns to train the backdoor policy and evaluate detection efficacy using a correspondingly updated GP model. For the Breakout game, we find that the updated GP model still achieves 0.837 AUROC in detection performance. We attribute this robustness to two factors. First, the optimized trigger pattern tends to overfit to the original GP model, preventing the adversarial property from transferring to the updated GP. Second, `PolicyGuard` takes both observations and actions as inputs; even if the observations can deceive the GP model, the triggered actions dynamically alter subsequent observations, causing the trajectory to deviate from the learned behavioral manifold and appear as outliers in the latent space. This cascading effect ultimately leads to the failure of the adaptive attack.

