# OpenReview forum: "PolicyGuard: Towards Test-time and Step-level Adversary Defense for Reinforcement Learning Agent"
_ICML.cc/2026/Conference — ICML 2026 regular_

### Official Review · Reviewer_Ercw · 2026-02-18

**Soundness:** 3
**Presentation:** 2
**Significance:** 3
**Originality:** 3
**Overall Recommendation:** 4
**Confidence:** 4

**Summary:**

This paper studies the problem of defending against backdoor attacks in RL. The authors consider two types of attacks, perturbation-based attacks, where the backdoors are injected into states, and adversarial-agent attacks, where the actions of an adversary trigger the backdoor to activate. They propose a defense method, PolicyGuard, which is suited for trained models deployed at test time. Importantly, their defense does not need access to model weights or full trajectory information. PolicyGuard uses collected trajectories to construct a GP model that captures inter-step and inter-episode correlations to construct a distribution on the state-action tuples. The intuition is that every suspicious state will yield high uncertainty (meaning that such occurrences are rare in the collected benign trajectories), which is quantified by the posterior variance of the model. In order to perform computation online, they utilize so called pseudo-trajectories, where they augment the missing trajectory at current step with already collected ones. They test their method on several benchmarks and compare with prior SOTA defenses. Their results show competitive performance across the board.

**Compliance With Llm Reviewing Policy:**

Affirmed.

**Final Justification:**

My questions have been resolved.

**Key Questions For Authors:**

1. Right after Equation (11), you say, given a suspicious state-action... How do we know whether that pair is suspicious before computing the posterior variance? Or do you construct pseudo-trajectories for every encountered state-action, and then compute the uncertainty?
2. Once you decide that a state-action is suspicious, what action does the agent then take at that state? A random action? Or one that minimizes the uncertainty. What I mean is, presuming that the backdoor is effective against the trained policy, how do you act when in triggered states?
3. How is the collection of the reference trajectories in the test environment done? Are you assuming a benign test environment first, where you can collect the data to build your model, and then you can deploy on actual attacked environment? Otherwise, how would you gather the clean data to build your model?
4. How many such trajectories are needed? I can see in the experiments that you collect 20000 trajectories of length 200. But it is not clear what the relationship is between the guarantee of the model and the number of collected trajectories. We know that in the limit the posterior variance goes to zero. Is there a theoretical quantification on how the data size scales with posterior variance?

**Limitations:**

Limitations are not provided in the paper. I would encourage the authors to clearly state them.

**Strengths And Weaknesses:**

**Strengths**

1. The results seem to be technically sound (although haven't checked the appendix proofs in detail), and experiments are well-conducted, with ablations on several directions.
2. What makes PolicyGuard significant is its competitive performance on original attacks and superior performance on hard-cded attacks, while not requiring any information on model parameters or full trajectory realization. At the same time, this defense generalizes well from perturbation-based to adversarial-agent attacks.
3. It seems to me that the idea of utilizing GPs on modeling the state distribution and then utilizing the posterior variance to detect anomalies in the current trajectory at test time, is fairly original.

**Weaknesses**

1. Although, on an intuitive level, the presentation of the used ideas is good and, in a way, the narrative proceeds smoothly, the technical presentation might need some improvement, and, at the same time, the paper contains many typos. There are many quantities that, although well-known to the expert reader, do need formal definitions and presentations for self-sufficiency of the paper. There are several such points, including Sections 2.1, 2.2, 3.2 and 3.3, where, for instance, it is not clear what is meant with $R(\pi(a|s;\theta),s)$ (although one can guess), the multiplication $s\odot m$, what $\gamma_t$ or $\gamma_e$ are, or how the matrices $K_{\cdot,\cdot}$ are defined. I believe the paper would improve a lot more if these vague notations were more explicit and properly defined, and typos (eg. in Section 2.1, do you mean transition instead of transaction?!) were fixed.

---

> ### Author Rebuttal · Authors · 2026-03-31
>
> We thank Reviewer Ercw for careful reviewing and thoughtful comments.
>
> ---
>
> **Q1: Notation and presentation issues.**
>
> We will fix all identified issues:
>
> - "transaction" → "transition" throughout
> -  We will fix Eq.(1) to use $R(s_t, a_t, s_{t+1})$ consistently with the MDP definition in Section 2.1.
> -  $s\odot m$ indicates observation $x$ point-wise multiple mask $m$.
> -  $k_{\gamma_t}(h_t^{(i)}, h_k^{(j)})$: the **step-level kernel** with lengthscale $\gamma_t$, which measures the similarity between hidden states across different episodes and timesteps, capturing inter-step temporal correlations. $k_{\gamma_e}(e^{(i)}, e^{(j)})$: the **episode-level kernel** with lengthscale $\gamma_e$, which measures the similarity between episodic embeddings across different episodes, capturing episode-level behavioral patterns.
> - We note that $h_t^{(i)}$, $e^{(i)}$ are defined in Section 3.2 (as RNN hidden states and episodic embeddings respectively). However, we acknowledge these definitions are scattered and not sufficiently prominent. We will consolidate all notation into a clear definition block at the beginning of Section 3.2 and add explicit element-wise definitions for kernel matrices: $K_{XZ}$ is a matrix (N×M) (e.g., $\[K_{XZ}\]{ij}=k_\gamma(x_i,z_j)$) to improve readability.
> ---
> **Q2: How do we know a pair is "suspicious" before computing variance?**
>  We apologize for the unclear wording.
>  - The term "suspicious" refers to any pair encountered in the potentially compromised environment. **We treat each incoming step's state-action pair as "suspicious" and compute posterior variance for every incoming state-action pair during deployment**
>  - We construct pseudo-trajectories for every encountered state-action, and then compute the uncertainty.
>
> ---
>
> **Q3: What action does the agent take after flagging?**
>
> In our current framework, we focus on detection. Upon flagging, practical and straightforward intervention strategies include:
>
> - execute a safe default action (e.g., no-op)
> - halt the agent and alert the human operator
> - query a redundant model from a different provider to predict the action
> - leverage existing methods such as SHINE[1] to recover the backdoor-infected observation and produce a benign action.
>
> ---
>
> **Q4: How are reference trajectories collected?**
>
> Yes — we assume a benign test environment is available first for collecting reference trajectories, **consistent with prior work[1,2,3]**. The defender collects clean trajectories, trains the GP, and **then deploys the defense in the potentially compromised environment.** We will make this two-phase procedure more explicit.
>
> ---
>
> **Q5: Data size.**
>
> We apologize for the insufficient information.
>
> - Our ablation in Appendix G.4 (Figure 11) shows AUROC as a function of training size (5K–30K), with performance plateauing around 20K, with reasonable performance at 15K. We will move this to the main paper.
> - Regarding theoretical scaling: from Corollary 3.2, posterior variance converges to zero as $N\rightarrow\infty$.
> - The convergence rate depends on data density and kernel lengthscale γ. We will take the theoretical analysis on the convergence rate for GP model in our future work.
>
> ---
>
> **Q6: Limitations.**
>
> **We will include a dedicated limitations section in our revision**:
> - **Clean Environment Assumption.** PolicyGuard requires access to a clean environment for collecting reference trajectories and training the GP model. While this assumption is standard across existing RL backdoor defenses (PolicyCleanse, SHINE, BIRD), it may not always be feasible in real-world scenarios where the deployment environment differs significantly from the test environment. Bridging this gap through domain adaptation or sim-to-real transfer techniques is an important direction for future work.
> - **Detection Only.** PolicyGuard focuses on detecting backdoor-infected steps and does not include a built-in intervention or mitigation mechanism. Upon detection, the system can be composed with existing input-level mitigation approaches (e.g., SHINE) or simple intervention strategies such as executing safe default actions. Developing a unified detect-intervene-mitigate pipeline remains future work.
> - **Gap between simulation and real-world.** Our evaluation is conducted entirely in simulated environments (Atari, MuJoCo). Real-world RL deployments (e.g., robotics, autonomous driving) involve noisy sensor inputs, partial observability, non-stationary dynamics, and physical constraints that are not fully captured by simulators. Whether the GP posterior variance remains a reliable uncertainty signal under these real-world conditions has not been validated. Developing robust uncertainty quantification methods that maintain detection efficacy across the sim-to-real transfer gap is a critical challenge for deploying PolicyGuard in practice.
>
> **References**
> 1. Yuan et al. SHINE. ICML'24.
> 2. Guo et al. PolicyCleanse. ICCV'23.
> 3. Chen et al. BIRD. NeurIPS'23.

---

> > ### Author Rebuttal · Reviewer_Ercw · 2026-04-01
> >
> > My questions have been resolved. I maintain my score.

---

### Official Review · Reviewer_D2ss · 2026-02-27

**Soundness:** 4
**Presentation:** 3
**Significance:** 3
**Originality:** 3
**Overall Recommendation:** 5
**Confidence:** 3

**Summary:**

This paper proposes a test-time, step-level backdoor defense for RL agents. The goal is to identify potentially triggered time steps online under a black-box setting. The key idea is to train an additive Gaussian Process (GP) trajectory model with an RNN-based encoder using benign trajectories collected from the suspicious agent in a “clean” environment. At deployment, the method uses the GP posterior variance as an uncertainty score: triggered steps are expected to deviate from the “normal behavior manifold,” leading to increased posterior variance. The paper reports that on seven environments, its AUROC outperforms prior defenses.

**Compliance With Llm Reviewing Policy:**

Affirmed.

**Final Justification:**

The authors’ response has addressed my concerns, and I believe this paper is worthy of acceptance.

**Key Questions For Authors:**

Please refer to the points raised in the **Weaknesses** above.

**Limitations:**

Insufficient. How to migrate the method from the simulation environment to real-world applications should be further discussed.

**Strengths And Weaknesses:**

**Strengths**

1. The paper is well written, with a clear and coherent methodological narrative.

2. Test-time defenses against RL backdoors are indeed important for safety-critical systems.

3. Using GP posterior variance as an uncertainty score for backdoor detection, combined with pseudo trajectories, is a reasonably novel and interesting design combination.

**Weaknesses**

1. The hard-coded setting appears to deviate from the standard RL backdoor threat model and may amount to “changing the problem.” The motivation and realism of this setting should be explicitly justified in the main text.

2. The pseudo trajectories mechanism may introduce distribution shift. The paper would benefit from a deeper analysis of the distributional similarity (or fidelity) between pseudo trajectories and true trajectories.

3. Fitting the GP requires the defender to have access to a clean environment, and the paper reports collecting 20,000 episodes. It is unclear how this dataset size is determined across environments and how performance scales with the number of episodes. Additional analysis (e.g., data-size sensitivity) would strengthen the paper’s soundness.

---

> ### Author Rebuttal · Authors · 2026-03-31
>
> We thank Reviewer D2ss for the careful reading and for recognizing our paper as "well written, with a clear and coherent methodological narrative," and the GP posterior variance approach as "a reasonably novel and interesting design combination."
>
> ---
>
> **Q1: Hard-coded attack deviates from standard RL backdoor threat model.**
>
> Thanks for your pointing out. We are deeply sorry we fail to provide sufficient information, which we want to clarify here.
>
> - From the attacker's perspective, we agree that the hard-coded attack differs from previous attacks in terms of attacker capability — it does not require any training or optimization. **However, from the defender's perspective, hard-coded attacks present the same threat model as standard attacks under black-box settings**: the defender faces an identical challenge of detecting anomalous behaviors from an agent whose internals are inaccessible, with the same attack objective of causing task failure upon trigger activation.
> - Our motivation for evaluating hard-coded attacks is twofold. **First, hard-coded attacks are arguably more realistic than standard learned backdoors in black-box deployment scenarios.** An adversary distributing a pre-trained model can trivially implement an if-conditional backdoor without any optimization trace — this represents the most natural attack vector for supply-chain threats and can be more easily deployed by attackers in practice. **Second, hard-coded attacks completely bypass all existing optimization-based defenses (NC[1], STRIP[3], SCALE-UP[4], PolicyCleanse[2] all yield AUROC ≈ 0.5), revealing a critical blind spot in current evaluation practices.** This demonstrates that including hard-coded attacks is essential for comprehensively assessing the robustness of any defense method intended for real-world deployment.
> - **We will include and detail the motivation in the main text.**
>
> ---
>
> **Q2: Pseudo trajectories may introduce distribution shift.**
>
> We are deeply sorry we fail to provide sufficient information, which we want to clarify here.
> - **In our additive GP model, pseudo trajectories maintain distributional fidelity for benign cases**: our additive GP model consists of step-level kernel $k_{\gamma_t}$ and episode-level kernel $k_{\gamma_e}$. The pseudo trajectories are essential for computing $k_{\gamma_e}$, which mainly depends on the episodic embedding $e^{(i)}$ derived from the terminal hidden state $h_T$. For benign policies, the distribution of $h_T$ remains stable across different pseudo trajectories, since the episode outcomes (win/loss or final scores) are distributionally consistent regardless of which clean reference trajectory provides the future context. This means $k_{\gamma_e}$ is largely preserved across pseudo trajectories for benign steps. Even though individual pseudo trajectories may introduce some distributional shift from the reference future context, our Interquartile Mean (IQM) aggregation (Eq. 13) effectively smooths out these fluctuations, keeping the uncertainty scores stably concentrated on the benign manifold.
> - However, for backdoor-infected policies, the triggered behavior at step t fundamentally disrupts the trajectory dynamics, causing $h_T$ to diverge from the benign manifold and resulting in consistently elevated posterior variance across all pseudo trajectories — regardless of which reference context is used. Since the anomaly originates from the substituted step itself, IQM aggregation cannot smooth away this signal, preserving a clear separation between benign and backdoor-infected steps. This is empirically validated by the clear distributional separation shown in Figure 2 and Figure 10.
>
> ---
>
> **Q3: 20K episodes — how determined, and data-size sensitivity.**
>
> - Our ablation in Appendix G.4 (Figure 11) shows AUROC as a function of training size (5K–30K), with performance plateauing around 20K, with reasonable performance at 15K. We will move this to the main paper.
> - Regarding theoretical scaling: from Corollary 3.2, posterior variance converges to zero as $N\rightarrow\infty$.
>
> ---
>
> **Q4: Limitation: Sim-to-real migration.**
>
> Thank you for this important suggestion.
> - **Sim-to-real migration for RL backdoor defenses is indeed a challenging and open problem, requiring careful consideration of noisy sensor inputs, partial observability, and non-stationary dynamics in real-world deployments.** To the best of our knowledge, no existing RL backdoor defense has been evaluated or deployed in real-world settings — this remains an important open direction for the entire field.
> - **We will pursue this in future work and add a dedicated limitations section — please see Q9 in Reviewer EufK**
>
> ---
> **References**
> 1. Wang et al. Neural Cleanse. IEEE S&P'19.
> 2. Guo et al. PolicyCleanse. ICCV'23.
> 3. Gao et al. STRIP. ACSAC'19.
> 4. Guo et al. SCALE-UP. ICLR'23.

---

> > ### Author Rebuttal · Reviewer_D2ss · 2026-04-01
> >
> > I appreciate the response and effort in addressing the review concerns. Overall, I believe this is a good paper worthy of acceptance, so I raise my score accordingly.

---

> > > ### Author Response · Authors · 2026-04-01
> > >
> > > We thank Reviewer D2ss for the careful reading and for recognizing our paper as **well written, with a clear and coherent methodological narrative,** and the GP posterior variance approach as **a reasonably novel and interesting design combination.**

---

### Official Review · Reviewer_EufK · 2026-03-06

**Soundness:** 3
**Presentation:** 3
**Significance:** 3
**Originality:** 3
**Overall Recommendation:** 4
**Confidence:** 3

**Summary:**

This paper introduces PolicyGuard, a test-time, step-level backdoor defense for black-box reinforcement learning (RL) agents. Existing defenses are severely limited by their need for white-box access and coarse-grained detection. PolicyGuard overcomes these issues by training an additive Gaussian Process (GP) on clean trajectories to capture benign behavior. It employs a novel pseudo-trajectory mechanism to compute GP posterior variance, quantifying the agent's epistemic uncertainty to detect backdoor-infected steps online. Supported by comprehensive theoretical proofs, experiments across various single- and multi-agent environments show PolicyGuard achieves state-of-the-art performance. Crucially, it demonstrates robust defense capabilities even against hard-coded attacks that bypass traditional optimization-based methods, ensuring reliable RL deployment in safety-critical scenarios.

**Compliance With Llm Reviewing Policy:**

Affirmed.

**Final Justification:**

The additional experiment and justificaition furtuer supported the proposed method, which raise my confidence of this method.

**Key Questions For Authors:**

1. Outdated and Insufficient Baselines
The defense baselines selected for benchmarking are relatively dated, with two approaches originating as far back as 2019. The empirical evaluation notably lacks comparisons with contemporary state-of-the-art (SOTA) defense mechanisms, which compromises the fairness and comprehensiveness of the performance assessment. Furthermore, the evaluated attack methods lack sufficient diversity, undermining the overall rigor of the robustness evaluation.

2. Ambiguous Latency Analysis
The time performance analysis lacks necessary granularity. While the manuscript claims a 9% latency improvement over the SHINE baseline, it fails to specify the exact scope of this metric. The authors must clarify whether this reported superiority reflects the end-to-end overhead of the complete defense pipeline or merely the single-step inference latency.

3. Distinguishing Backdoors from Natural Anomalies
Currently, the trajectories collected for model training are derived exclusively from benign data. However, real-world deployment scenarios inevitably involve non-malicious anomalous data (e.g., extreme outliers or naturally occurring out-of-distribution samples) alongside standard benign and backdoor data. How does the proposed framework ensure it can reliably distinguish genuine backdoor attack samples from non-malicious anomalies without triggering false positives?

**Limitations:**

yes

**Strengths And Weaknesses:**

1. Soundness
The paper features a rigorous theoretical framework and standardized experiments across 7 environments, effectively validating the pseudo-trajectory mechanism. However, its reliance on strong assumptions and large-scale clean trajectories is impractical for real-world deployments. Furthermore, the evaluation omits stealthy/clean-label attacks and relies solely on AUROC, neglecting critical security metrics like False Positive Rate and Recall.

2. Presentation
The manuscript is well-structured, visually clear, and highly reproducible with a meticulously detailed appendix. However, it fails to sufficiently differentiate the core pseudo-trajectory mechanism from existing temporal encoding techniques. Additionally, the paper lacks a systematic discussion of its limitations and contains minor mathematical notation inconsistencies.

3. Significance and Originality
This work pioneers online, step-level backdoor detection for black-box RL agents, demonstrating exceptional robustness against hard-coded attacks. While highly valuable for safety-critical deployments, its algorithmic originality is somewhat constrained by its reliance on established OOD paradigms. Furthermore, the framework lacks cross-policy transferability and only addresses the detection phase, falling short of a complete "detect-intervene-mitigate" defense pipeline.

---

> ### Author Rebuttal · Authors · 2026-03-31
>
> Dear Reviewer Eufk, thanks for your careful review of our paper and thoughtful comments.
>
> ---
> **Q1: Outdated and insufficient baselines.**
>  - **Defense:** Our 6 baselines cover supervised-learning-adapted[2,7,4,6] and RL-specific defenses (PolicyCleanse[3], SHINE[1]), following SHINE[1]'s standard. **We emphasize that our comparison includes the two most recent and competitive RL-specific defenses, SHINE[1] and PolicyCleanse[3];** and PolicyGuard outperforms both. **The adapted SL baselines[2,7,4,6] serve to demonstrate that SL-originated defenses are insufficient for RL backdoor detection,** particularly under hard-coded attacks. **Other recent RL defenses[8,5] require white-box access or different threat models.** We welcome suggestions of methods applicable to our setting.
>  - **Attacks:** We evaluate against Q-Inception[9] following  https://github.com/EthanRath/Backdoors-In-RL., the most recent RL backdoor attack (4x4 trigger, ~40M steps):
>  | Metric | Pong | Breakout |
>    |:---|:---|:---|
>    | AUROC | .843 | .864 |
>    | FPR | .101 | .097 |
>    | Recall | .753 | .796 |
>  - **Our novel hard-coded attack — not in any prior work** — bypasses all optimization-based defenses, broadening evaluation beyond existing benchmarks.
>
> ---
> **Q2: Ambiguous latency analysis.**
> Fig.(3) reports **end-to-end per-step inference latency**: RNN encoding, GP variance computation, IQM aggregation. We will clarify in revision.
>
> ---
> **Q3: Lack of evaluation metrics.**
>  we set threshold sas 90th percentile of clean uncertainty distribution, bounding FPR at ~10%:
>  | Metric | Pong | Brkout | SpInv | RTGA | RTGH | YSNP | Sumo |
> |:---|:---|:---|:---|:---|:---|:---|:---|
> | AUROC | .857 | .869 | .842 | .766 | .895 | .896 | .881 |
> | FPR | .101 | .098 | .102 | .099 | .101 | .098 | .100 |
> | Recall | .785 | .806 | .751 | .605 | .861 | .863 | .832 |
>  Hard-coded:
>
>  | Metric | Pong | Breakout | Spinv | RTGA | RTGH | YSNP | Sumo |
> | :--- | :--- | :--- | :--- | :--- | :--- | :--- | :--- |
> | **AUROC** | 0.861 | 0.878 | 0.864 | 0.827 | 0.904 | 0.901 | 0.881 |
> | **FPR** | 0.099 | 0.102 | 0.097 | 0.101 | 0.099 | 0.102 | 0.098 |
> | **Recall** | 0.792 | 0.825 | 0.796 | 0.723 | 0.874 | 0.871 | 0.832 |
>
> Full TPR-FPR tradeoff analysis will be included in revision.
>
> ---
> **Q4: Distinguishing backdoors from natural anomalies.**
>  - **No anomaly-based defense is completely immune to FP on rare data, is inherent to all RL backdoor detection**, not specific to PolicyGuard.
> - **Our GP is trained with different random seeds per episode and stochastic action sampling (not deterministic argmax).** Competitive agents achieve ~50% win rates, so training data covers wins, losses, and edge cases from multi-agent stochasticity.
> - Our average AUROC of 0.856 confirms effective separation between backdoor and benign behaviors. Even if rare benign states trigger occasional false positives, their rarity makes the impact negligible. Our primary goal is filtering backdoor-infected steps, which is not compromised by sporadic FPs on extreme edge cases.
> - Our ablation on random perturbations (Fig. 4(d)) shows robust performance under perturbation magnitudes up to 20% —PolicyGuard does not conflate natural deviations with backdoor behaviors.
>
> ---
> **Q5: Reliance on large-scale clean trajectories.**
>  - Standard assumption cosnistent with previous work[1,3,5].
> - Previous work[1,3] requires equal or more trajectories. Ablation (Appendix G.4) shows performance plateaus at 20K. Will move to main paper.
>
> ---
> **Q6: Cross-policy transferability.**
>  - **No existing RL defense supports cross-policy transfer[1,3,5]**
> - **This is inherently challenging: different games have fundamentally different state-action distributions and attack types (perturbation-based vs. adversary-agent).** We will discuss this as a limitation&future direction.
>
> ---
> **Q7: Only detection, no intervention/mitigation.**
>  - **Detecting backdoor inputs for black-box models is a well-established research problem[7,8].**
> - **See Q3 for Reviewer Ercw.**
> - Under black-box settings, parameter-level mitigation is infeasible by definition; our detection composes naturally with input-level mitigation (e.g., SHINE[1]). We will discuss this pipeline in the revision.
>
> ---
> **Q8: Pseudo-trajectory vs temporal encoding.**
> Temporal encoding provides ordering info but cannot compensate for **missing future steps**. Our GP requires $h_T$ from complete trajectories for episodic embedding $e^{(i)}$. Pseudo trajectories fill this gap.
>
> ---
> **Q9: Limitations.**
> We will include a limitations section in the revision, see our response to Reviewer Ercw (Q6). Happy to expand further per reviewer feedback.
>
> ---
> References
> 1. Yuan et al. SHINE. ICML'24.
> 2. Wang et al. Neural Cleanse. IEEE S&P'19.
> 3. Guo et al. PolicyCleanse. ICCV'23.
> 4. Dong et al. B3D. ICCV'21.
> 5. Chen et al. BIRD. NeurIPS'23.
> 6. Gao et al. STRIP. ACSAC'19.
> 7. Guo et al. SCALE-UP. ICLR'23.
> 8. Bharti et al. Provable Defense. NeurIPS'22.
> 9. Rathbun et al. Q-Inception. ICML'25.

---

> > ### Author Rebuttal · Reviewer_EufK · 2026-04-01
> >
> > My concerns have been adequately addressed.

---

> > > ### Author Response · Authors · 2026-04-01
> > >
> > > We thank Reviewer EufK for the thorough evaluation, and for recognizing that our work **pioneers online, step-level backdoor detection for black-box RL agents** with **exceptional robustness against hard-coded attacks** and is **highly valuable for safety-critical deployments.**
> > >
> > > We will include the analyses of TP-FP for our approach in our revision.
> > >
> > > **We discuss several limitations of our work to provide a complete picture of PolicyGuard's applicability and to motivate future research directions. We will include this in our revision.**
> > >
> > > - **Clean Environment Assumption.** PolicyGuard requires access to a clean environment for collecting reference trajectories and training the GP model. While this assumption is standard across existing RL backdoor defenses (PolicyCleanse, SHINE, BIRD), it may not always be feasible in real-world scenarios where the deployment environment differs significantly from the test environment. Bridging this gap through domain adaptation or sim-to-real transfer techniques is an important direction for future work.
> > > - **Detection Only.** PolicyGuard focuses on detecting backdoor-infected steps and does not include a built-in intervention or mitigation mechanism. Upon detection, the system can be composed with existing input-level mitigation approaches (e.g., SHINE) or simple intervention strategies such as executing safe default actions. Developing a unified detect-intervene-mitigate pipeline remains future work.
> > > - **Gap between simulation and real-world.** Our evaluation is conducted entirely in simulated environments (Atari, MuJoCo). Real-world RL deployments (e.g., robotics, autonomous driving) involve noisy sensor inputs, partial observability, non-stationary dynamics, and physical constraints that are not fully captured by simulators. Whether the GP posterior variance remains a reliable uncertainty signal under these real-world conditions has not been validated. Developing robust uncertainty quantification methods that maintain detection efficacy across the sim-to-real transfer gap is a critical challenge for deploying PolicyGuard in practice.

---

### Official Review · Reviewer_Na9A · 2026-03-10

**Soundness:** 3
**Presentation:** 3
**Significance:** 3
**Originality:** 3
**Overall Recommendation:** 4
**Confidence:** 2

**Summary:**

This paper introduces PolicyGuard, a test-time and step-level defense against backdoor attacks in reinforcement learning. By leveraging Gaussian Process posterior variance and adapting pseudo-trajectories, the method estimates uncertainty at individual time steps. This allows PolicyGuard to detect potentially compromised steps online without requiring access to the agent's internal parameters or full episode trajectories.

**Compliance With Llm Reviewing Policy:**

Affirmed.

**Final Justification:**

The authors further addressed my primary concerns regarding the computational complexity and the soundness of evaluation. I will keep my positive score.

**Key Questions For Authors:**

1.	What is the computational complexity and operational overhead of the proposed PolicyGuard method?
2.	Can the authors discuss potential adaptive attacks or strategies an adversary might use to bypass PolicyGuard?

**Limitations:**

yes

**Strengths And Weaknesses:**

Strengths:

1.	Defending against backdoor attacks is critical for the safety and robustness of reinforcement learning systems. This work represents a valuable step forward in this important research area.
2.	The presentation of this paper is overall well-organized.
3.	The majority of claims are well-supported.

Weaknesses:

1.	Some mathematical notations are not rigorous and lack consistency. For example, in Section 2.1, the reward function $R$ is defined as $\mathcal{S} \times \mathcal{A} \times \mathcal{S} \rightarrow \mathbb{R}$. However, Equation (1) uses it as $R(\pi(a_t \vert s_t; \theta), s_t)$, which does not align with the formal definition.
2.	In section 2.1, the transaction function is defined as deterministic. Since this is a restricted special case of a general Markov Decision Process (MDP), it is currently unclear whether the results can be directly extended to environments with stochastic transitions.
3.	The empirical evaluation relies on older backdoor attacks [1, 2]. Incorporating more recent and advanced attacks, such as [3], would significantly strengthen the soundness of the evaluation and demonstrate the true robustness of the defense.

[1] Kiourti et al. Trojdrl: evaluation of backdoor attacks on deep reinforcement learning. DAC 2020.

[2] Wang et al. Backdoorl: Backdoor attack against competitive reinforcement learning. IJCAI 2021.

[3] Rathbun et al. Adversarial Inception Backdoor Attacks against Reinforcement Learning. ICML 2025.

---

> ### Author Rebuttal · Authors · 2026-03-31
>
> ---
>
> Dear Reviewer Na9A, thank you very much for your careful review of our paper and thoughtful comments.
>
> ---
> **Q1**: Notation inconsistency (R defined as $S×A×S→ℝ$ but Eq.(1) uses different form).
>
> **R1**: Thank you for catching this and sorry for causing you confused. We will fix Eq.(1) to use $R(s_t, a_t, s_{t+1})$ consistently with the MDP definition in Section 2.1. The current form was an informal shorthand that sacrificed rigor.
>
> ---
> **Q2**: Deterministic transition assumption — extensibility to stochastic MDPs.
>
> **R2**: Thank you for these insightful comments! We hope the following explanations can further alleviate your concerns.
> - Our framework does not fundamentally require deterministic transitions. The GP model is trained on observed trajectories, which naturally reflect any environment stochasticity. The posterior variance captures deviation from the learned behavioral manifold regardless of transition dynamics — the GP operates on the empirical trajectory distribution rather than requiring knowledge of the transition kernel. The deterministic formulation in Section 2.1 was adopted following previous widely-adopted work's settings.
> - We note that all existing related RL backdoor attacks and defenses(TrojDRL[7], BackdooRL[8], PolicyCleanse[4], SHINE[2], etc) are evaluated on the same set of widely-adopted benchmark environments (Atari, MuJoCo competitive games) under deterministic transition settings. Our experimental setup follows this established convention to ensure fair and consistent comparison. We will add a remark clarifying PolicyGuard's applicability to stochastic MDPs and consider extending evaluation to stochastic environments in future work.
>
>
>
> ---
> **Q3**: Should include Rathbun et al. (ICML 2025)[1].
>
> **R3**:  Thanks for your comments.
> We evaluate against the most recent backdoor attacks against RL: Q-Inception following  https://github.com/EthanRath/Backdoors-In-RL. As Q-Inception is designed as the Perturbation-based attack, we follow the same settings for trigger patterns as for TrojDRL (i.e., 4x4 random square). We evaluate PolicyGuard against Q-Inception on two Atari games (i.e., Pong and Breakout.) We run ~40M time steps for each to ensure it achieves the optimal attack efficacy. The results show as below:
>    | Metric | Pong | Breakout |
>    | :--- | :--- | :--- |
>    | **AUROC** | 0.843 | 0.864 |
>    | **FPR** | 0.101 | 0.097 |
>    | **Recall** | 0.753 | 0.796 |
>
> We will include and cite Rathbun et al.[1] and its comparison results into our revision.
>
> ---
> **Q4**: Computational complexity of PolicyGuard.
>
> **R4**: Thank you for pointing it out! We are deeply sorry that our submission failed to provide sufficient inference complexity that we want to clarify here. PolicyGuard's inference phase contains **GP posterior variance computation (Eq.7)** and **RNN forward pass**.
> - **GP posterior variance computation (Eq.7):** The per-step GP posterior variance computation is $O(M^{2})$ where M is the number of inducing points ($i.e.,$ 100 and 600 for Atari and Mujoco games) during the inference phase, since $K_{ZZ}^{-1}$ is precomputed during training. The additional cost per step is the RNN forward pass for pseudo trajectory encoding.
> - **RNN forward pass:** At step t, doing one incremental RNN step ($h_t = \text{GRU}(x_t, h_{t-1})$), that's $O(q^{2})$ where q is hidden dim.
> - We show the latency for implementing PolicyGuard and comparing to previous work in Fig.(3). PolicyGuard would cause around 0.08s for each step in our experimental settings.
>
> ---
> **Q5**: Adaptive attacks to bypass PolicyGuard.
>
> **R5**: Thanks for pointing that, we are sorry to make you misunderstand that we want to clarify. We addressed this in **Section 4.3 and Appendix G.6**. **Our key finding is a fundamental tradeoff: adversaries cannot simultaneously maintain attack efficacy, clean performance, and low uncertainty.** For adversary-agent adaptive attacks, trojans fail to activate at default poisoning rates (≤40%); increasing the rate degrades clean performance while raising uncertainty U. For perturbation-based adaptive attacks (Appendix G.6), the optimized trigger overfits to the original GP and does not transfer to the updated GP. Furthermore, even if observations deceive the GP, the triggered actions dynamically alter subsequent observations, causing trajectory deviation. **We will highlight these results more prominently in the main text.**
>
> References
>
> 1. Rathbun et al. Adversarial Inception Backdoor Attacks against Reinforcement Learning. ICML, 2025.
> 2. Yuan et al. SHINE: Shielding Backdoors in Deep Reinforcement Learning. ICML, 2024.
> 3. Guo et al. PolicyCleanse: Backdoor Detection and Mitigation for Competitive Reinforcement Learning. ICCV, 2023.
> 4. Wang et al. BackdooRL: Backdoor Attack against Competitive Reinforcement Learning. IJCAI, 2021.
> 5. Kiourti et al. TrojDRL: Evaluation of Backdoor Attacks on Deep Reinforcement Learning. DAC, 2020.

---

> > ### Author Rebuttal · Reviewer_Na9A · 2026-04-01
> >
> > I would like to thank the authors for their responses for addressing my concerns. I will keep my positive score.

---

> > > ### Author Response · Authors · 2026-04-01
> > >
> > > Dear Reviewer Na9A, thank you very much for your careful review of our paper and thoughtful comments. We are encouraged by your positive comments on our **a valuable step forward in this important research area**, **presentation is well-ornganized**, and **the majority of claims are well-supported**.
> > >
> > > We will fix all the typos and notation incosistency in our revision, follows:
> > >
> > > - "transaction" → "transition" throughout
> > > - We will fix Eq.(1) to use $R(s_t, a_t, s_{t+1})$ consistently with the MDP definition in Section 2.1.
> > > - $s\odot m$ indicates observation $x$ point-wise multiple mask $m$.
> > > $k_{\gamma_t}(h_t^{(i)}, h_k^{(j)})$: the step-level kernel with lengthscale $\gamma_t$, which measures the similarity between hidden states across different episodes and timesteps, capturing inter-step temporal correlations. $k_{\gamma_e}(e^{(i)}, e^{(j)})$: the episode-level kernel with lengthscale $\gamma_e$, which measures the similarity between episodic embeddings across different episodes, capturing episode-level behavioral patterns.
> > > - We note that $h_t^{(i)}$, $e^{(i)}$ are defined in Section 3.2 (as RNN hidden states and episodic embeddings respectively). However, we acknowledge these definitions are scattered and not sufficiently prominent. We will consolidate all notation into a clear definition block at the beginning of Section 3.2 and add explicit element-wise definitions for kernel matrices: $K_{XZ}$ is a matrix (N×M) (e.g., $[K_{XZ}]{ij}=k_\gamma(x_i,z_j)$) to improve readability.

---

### Decision · Program_Chairs · 2026-04-30

**Decision:**

Accept (regular)

**Comment:**

Great paper, as all reviewers agree and I share the positive opinion.